



# An Extended Kriging method to interpolate soil moisture data measured by wireless sensor network

Jialin Zhang [1, 2], Xiuhong Li [1, 2], Qiang Liu [1, 2], Long Zhao [1, 2], Baocheng Dou [1, 2]

[1] College of Global Change and Earth System Science, Beijing Normal University, Beijing 100875, China

[2] State Key Laboratory of Remote Sensing Science, Jointly Sponsored by the Institute of Remote Sensing and Digital Earth of Chinese Academy of Sciences and Beijing Normal University, Beijing 100101, China

*Correspondence to*: Qiang Liu (toliuqiang@bnu.edu.cn)

**Abstract.** In recent years, wireless sensor network (WSN) has emerged as a new technique to collect Earth observation data at a relatively low cost and minimal labor over large areas. However, WSN observations are still point data. To determine the

spatial distribution of a land surface parameter, interpolation of these point data is necessary. Some geostatistical interpolation methods, such as the Ordinary Kriging method, Co-Kriging method, and Regression Kriging method, have been used in various fields. However, capturing the spatial distribution pattern of heterogeneous land surface parameters is still difficult. For example, near-surface soil moisture is a critical parameter for agriculture management, and hydrological and ecological research. However, as soil moisture is related to many factors such as topography, soil type, and vegetation, even a WSN

observation grid is not sufficiently dense to reflect its spatial distribution pattern. This study developed a method to interpolate WSN-measured soil moisture with the aid of remote sensing images. The underlying idea is extension of the traditional Kriging algorithm by introducing spectral variables, specifically, vegetation index (VI) and albedo, from satellite imagery as supplementary information to aid interpolation. Thus, the new Extended Kriging algorithm operates on spatial and spectral combined space. The algorithm has been applied to WSN-measured data in the HiWATER campaign to generate daily soil

moisture maps in the 5 km × 5 km oasis area in the middle reaches of the Heihe River, western China, from June 10 to July 15, 2012. Visual inspections indicate that the result from the Extended Kriging algorithm shows more spatial details than that of the traditional Kriging algorithm, and the temporal variation of patch-average soil moisture is, in general, consistent with precipitation/irrigation data. Leave-one-out cross-validation was also adopted to estimate the interpolation accuracy. The Root Mean Square Error (RMSE) of the Extended Kriging method was also found to be smaller than that of the original Ordinary

Kriging method. Analysis with minimum variance of error ($\sigma_k$), a self-uncertainty indicator given by the Kriging algorithm, also gave the same conclusion. Further testing also indicated that if high-resolution land surface temperature maps are available, they can be added to the spectral variables and further improve the interpolation accuracy.

**Key words:** wireless sensor network, Kriging interpolation, soil moisture, spectral variables



## 1 Introduction

Wireless sensor network (WSN) is a novel technique for ground data collection that is currently in high demand. It has been applied in various fields, such as hydrology, soil environment, atmospheric environment, forest meteorology, and fire disaster (Othman and Shazali, 2012). WSN is a cross-discipline technology that integrates sensor, automatic control, communication,

and data analysis (Peng, 2007). In contrast to traditional ground data observation methods, the multiple sites comprising a WSN facilitate measurement of regional distribution of parameters. Further, its small size and relatively low price (Othman and Shazali, 2012) enables more nodes to be installed in the study area to meet research requirements. This helps us to obtain more extensive and much denser observation data of a specific land surface parameter. Compared with satellite remote sensing technique, WSN is a more flexible platform that can support a larger variety of sensors (Peng, 2007; Jianzhong and Hong,

2008). It can measure leaf area index (LAI), soil moisture, soil temperature, electrical conductivity, and atmospheric infrared radiance temperature at each observation node (Li et al., 2015; Dou et al., 2016). Furthermore, in contrast to remote sensing, in which data are obtained during overpass periods, monitoring by WSN is continuous and in real time. This complements the area data measured by remote sensing and also provides reference data for validation of satellite and aerial remote sensing (Li et al., 2016).

However, WSN only measures point data, whereas spatially continuous data over a certain region are increasingly required in environmental sciences and management in order to make effective decisions or justified interpretations (Li and Heap, 2011). Consequently, to determine the spatial distribution of a land surface parameter, data interpolation is necessary. Some traditional interpolation methods, such as the inverse distance weighting method (ISW), polynomial method, and Kriging method, have been widely used to convert point data to spatial distribution. The ISW method and the polynomial method are

types of non-geostatistical interpolation methods, which are relatively fast and easy to compute, but have estimation accuracies that are limited by their algorithms (Lu and Wong, 2008). By contrast, the Kriging method uses variogram analysis to estimate the spatial variation structure, and takes spatial autocorrelation into consideration (Aalto et al., 2013). The Kriging method is a group of stochastic interpolation methods comprising the Simple Kriging method, Ordinary Kriging method, Universal Kriging method, Co-Kriging method, Regression Kriging method, and Residual Kriging method (Oliver and Webster, 1990).

These methods have been used in various studies. For example, Maruyama et al. (2010) estimated peak ground velocities using the Simple Kriging method. Pokhrel et al. (2013) used the Simple Kriging method to estimate liquefaction potential and also to validate interpolation results. Wu et al. (2013) utilized the Residual Kriging method with the input variables of latitude, longitude, and elevation to estimate the average monthly temperature in the United States. Aznar et al. (2013) used the Co-Kriging method and the Canadian regional climate model as secondary information to spatially interpolate a mean monthly

temperature time series. Liang et al. (2016) used the Co-Kriging method to estimate daily $NO_3$-N loads in an agricultural river with the assistance of daily discharge. Through the development of geostatistical interpolation methods, more information about the rule or pattern of the distribution of the target parameter was explored, either through statistics of this parameter, or through its correlation with other parameters.



Soil moisture is a familiar yet vital factor in agriculture, ecology, and hydrological cycle studies (Dripps and Bradbury, 2007; Toth et al., 2008; Yu et al., 2011; Wang et al., 2012). Despite advances in interpolation algorithms, estimation of the spatial distribution of near-surface soil moisture is still unsatisfactory because soil moisture is highly heterogeneous on both spatial and temporal scales even over short distances (Gómez‑Plaza et al., 2000). Many researchers have applied remote

sensing to invert soil moisture. For example, Fan et al. (2015) improved the performance of Ts/VI space in retrieving soil moisture based on Compact Airborne Spectral Imager (CASI)/Thermal Airborne Spectrographic Imager (TASI) data. El Hajj et al. (2016) used neural networks (NNs) to estimate surface soil moisture from X-band SAR data over irrigated grassland areas. Ponnurangam et al. (2016) used a compact polarimetric decomposition with surface component inversion to estimate soil moisture on bare and vegetated agricultural soils. However, soil moisture imposes significant difficulty in quantitative

remote sensing inversion because, whereas optical remote sensing is not directly sensitive to soil moisture, thermal infrared and microwave remote sensing usually have low spatial resolution. Ground observation data, such as that from WSN measurement, are always necessary as supplementary data for remote sensing inversion of soil moisture. In other words, remote sensing information can always be viewed as supplementary data to aid the interpolation of ground-measured soil moisture. In this light, Yao et al. (2013) used four spatial interpolation methods to estimate soil moisture in a complex terrain catchment

and Regression Kriging proved to be optimal in producing a map with more details and better accuracy than other methods. Liao et al. (2016) analyzed various sources of uncertainty, such as soil properties and terrain indices, while estimating near-surface soil moisture content with the aid of Co-Kriging at two typical hill slopes. However, as soil moisture is correlated with many factors, such as elevation, vegetation, temperature, and irrigation, it is rather difficult to single out an optimally related factor to aid interpolation in the Regression Kriging or Co-Kriging algorithm. Some researchers have also tried to use data

assimilation methods in soil moisture estimation. Gao et al. (2014) estimated the spatial pattern of soil moisture using the BME method, which is based on WSN data and auxiliary information from ASTER (Terra) land surface temperature measurements.

In this paper, we propose extending the Kriging method by introducing high-resolution remote sensing imagery spectral variables into the interpolation algorithm. The spectral variables, such as vegetation index (VI), albedo, and temperature, can be either directly or indirectly relevant to soil moisture. The new Extended Kriging algorithm operates in the combined space

of spectral dimension and spatial dimension. Thus, the semivariogram is also estimated in the combined space. The proposed algorithm has been tested with soil moisture data acquired by WSN in the HiWATER campaign to generate a daily soil moisture map at 30 m spatial resolution and daily temporal resolution.

## 2  Study area and dataset

### 2.1  Study area

The study area, shown in Fig. 1, is located in Zhangye oasis in the middle reaches of the Heihe River Basin (HRB) in northwestern China (38.87 ˚N, 100.40 ˚E). The HRB is the second largest inland river basin and is characterized by large areas



of alpine cold and arid landscapes and a small portion of oasis agricultural land. The main oasis in the middle reaches is agricultural land with a variety of vegetation, including trees, maize, wheat, and vegetables. As the potential evaporation is very high, ranging from 1200 mm to 1800 mm per year, while the annual precipitation in the artificial oasis is only 177 mm, irrigation is the primary source of water for crops (Li et al., 2013).

The HRB has long served as a testbed for integrated watershed studies and hydrological experiments (Cheng, 2009). HiWATER is an ongoing watershed eco-hydrology comprehensive experiment that began in 2012. With the objective of improving the comprehensive observation ability, an eco-hydrology WSN was installed as a part of the HiWATER basic experiment. From May 2012 to September 2012, 50 WATERNET nodes were installed in a 5.5 km ×5.5 km forci experimental area in the main oasis in the middle reaches.

In this study, we chose an area approximately 4.5 km ×5.0 km covered by WATERNET as the experimental area, which consists of 48 WATERNET nodes (Fig. 1). All the nodes were located in a cornfield.

## 2.2   Data resource

WSN data and remote sensing data were both used in this study. We used the data covering the period from June 10, 2012 to July 15, 2012, when the vegetation cover of the study area changed from sparse to dense. WSN data were obtained using the

48 WATERNET soil moisture sensors in the experimental area. Each of the WATERNET nodes measured soil moisture (SM) and soil temperature in two layers (4 cm, 10 cm), and the data were collected every 5 minutes (Jin et al., 2014). The soil moisture measurements are based on the frequency-domain reflectometry method using a Hydro Probe II (HP-II) sensor (Gao et al., 2014). Some WATERNET design details and other information are given in Jin et al. (2012). Because the data collected by WSN are influenced by sensor noise and other abnormal conditions during data transfer, smoothness and noise reduction

treatments are necessary. We first excluded the abnormal data; then, averaged the data for the whole day, and then used the averaged result as the final soil moisture value for each node.

Remote sensing data served as auxiliary data in estimation of the distribution of near-surface soil moisture. NDVI and albedo were the spectral variables adopted to aid interpolation of the WSN-measured soil moisture. NDVI and albedo were derived from CCD camera onboard the Chinese HJ satellite. The spatial resolution was 30 m and revisiting frequency

approximately 2 days. Owing to the influence of clouds, only five clear-sky images on the following dates during this period could be used: June 15, June 19, June 29, July 8, and July 13 in the year 2012. Land surface temperature is also an important variable related to soil moisture. The data were acquired from the airborne sensors of CASI and TASI on July 10, 2012 (Fan et al., 2015). The spatial resolution was 2.5 m.



## 3 Method

The new spatial interpolation method proposed in this paper is based on the traditional Kriging algorithm. The proposed method extends the traditional X and Y spatial coordinates to spatial and spectral coordinates, and utilizes a vegetation index and albedo in the interpolation algorithm as supplementary information. In this section, first, an outline of the traditional Kriging algorithm is given, then the technique employed to extend the Kriging algorithm is explained. While fitting the semivariogram of the soil moisture, as the number of WSN nodes is insufficient to gather robust statistics, we used a remote sensing-derived soil moisture map to calculate the variance function.

### 3.1 Traditional Kriging method

#### 3.1.1 Basic formula

Kriging is an interpolation method derived from regionalized variable theory, which inherited the concept from geostatistics (Oliver and Webster, 1990). It has been used to provide linear unbiased predictions at unsampled locations and depends on expression of the spatial variation of the property in terms of the semivariogram (Burgess and Webster, 1980; Cressie, 1990). This method quantifies and reduces the uncertainties of estimation, minimizing self-estimated prediction errors (Gao et al., 2014). The core of Kriging is an optimally linear unbiased estimator that can be expressed as follows (Journel and Huijbregts, 1978):

$$Z^*(v_0) = \sum_{i=1}^{n} \lambda_i Z(v_i) \qquad (1)$$

Optimal estimation requires the minimum variance of errors:

$$\sigma_\kappa = \mathrm{Var}[Z(v_0) - Z^*(v_0)] = \mathrm{E}\left\{ \left[ Z(v_0) - \sum_{i=1}^{n} \lambda_i Z(v_i) \right]^2 \right\} = \min \qquad (2)$$

To ensure unbiased estimation, the following constraint must satisfy the equation as follows:

$$\sum_{i=1}^{n} \lambda_i = 1 \qquad (3)$$

To solve this constrained optimization problem, the Lagrange Multiplier Method (LMM) is adopted. With Eq. (2) as the objective function and Eq. (3) as the constraint, the LMM minimizes the following cost function:

$$f(\lambda_1, \lambda_2, \cdots, \lambda_n, \mu) = \frac{1}{2} \mathrm{E}\left\{ \left[ Z(v_0) - \sum_{i=1}^{n} \lambda_i Z(v_i) \right]^2 \right\} + \mu\left( 1 - \sum_{i=1}^{n} \lambda_i \right) \qquad (4)$$

where $\mu$ is the Lagrange Multiplier. At the minimum point of the cost function, the differentiation of $f$ with respect to each of its variables is zero. Thus, the optimization problem decomposes into one of solving the following set of equations:





$$\begin{cases} \dfrac{\partial f}{\partial \lambda_i} = 0, \quad i = 1, 2, \cdots, n \\ \dfrac{\partial f}{\partial \mu} = 0 \end{cases} \qquad (5)$$

Differentiating the cost function, we have

$$f(\lambda_1, \lambda_2, \cdots, \lambda_n, \mu) = \frac{1}{2} \mathrm{E}\left\{ \left[ Z(v_0) - \sum_{i=1}^{n} \lambda_i Z(v_i) \right]^2 \right\} + \mu \left( 1 - \sum_{i=1}^{n} \lambda_i \right) \qquad (6)$$

$$\begin{cases} \dfrac{\partial f}{\partial \lambda_i} = \lambda_i \mathrm{E}[Z^2(v_i)] + \displaystyle\sum_{\substack{j=1 \\ j \neq i}}^{n} \lambda_j \mathrm{E}[Z(v_i)Z(v_j)] - \mathrm{E}[Z(v_0)Z(v_i)] - \mu = 0 \\ \dfrac{\partial f}{\partial \mu} = 1 - \displaystyle\sum_{i=1}^{n} \lambda_i = 0 \end{cases} \qquad (7)$$

If we know that $\mathrm{E}[Z^2(v_i)]$, $\mathrm{E}[Z(v_0)Z(v_i)]$, and $\mathrm{E}[Z(v_i)Z(v_j)]$, then the equations can be solved. These values are estimated by the semivariogram function in Sect. 3.1.2.

    The minimum variance of error ($\sigma_k$), as is shown in Eq. (2), can be used as a quality indicator in estimation (Yamamoto, 2000). It can evaluate the uncertainty and the estimation accuracy from the algorithm itself.

**3.1.2  Estimating semivariance and semivariogram**

    Semivariance and semivariogram, containing spatial correlation information, are important concepts in geostatistics. The semivariance of variables at certain locations is estimated from the semivariogram function, which is a function of the distance between the two locations. Usually, a de-trending preprocess is applied to the observation data. After this preprocessing, the spatial distribution of the variable is assumed stationary, which means that the semivariance does not change with location.

On the basis of this assumption, the semivariance can be estimated from the data that a random variable is well correlated in space as a function of separation distance. The semivariance ($\gamma$) of Z between two data points is defined as

$$\gamma(x_i, x_0) = \gamma(h) = \frac{1}{2} \mathrm{Var}[Z(x_i) - Z(x_0)] \qquad (8)$$

where $h$ is the distance between points $x_i$ and $x_0$ and $\gamma(h)$ is the semivariogram (Webster and Oliver, 2001).

    The semivariogram is usually estimated from the statistics of sample points as follows:

$$\hat{\gamma}(h) = \frac{1}{2n} \sum_{i=1}^{n} \left( Z(x_i) - Z(x_i + h) \right)^2 \qquad (9)$$

where $n$ is the number of pairs of sample points separated by distance $h$ (Burrought and McDonnel, 1998).

    As the number of WSN nodes is insufficient to gather robust statistics, a soil moisture map was used here to derive the semivariogram function. This soil moisture map was derived from the airborne hyperspectral datasets of CASI/TASI (Fan et al., 2015), acquired on July 10, 2012, between 12:00 and 12:30 (local time), at an altitude of 2500 m. In the calculation, we





sampled 3000 points every time from the soil moisture map and calculate the semivariance by repeating the sampling thrice. The semivariance calculated from the soil moisture map is shown in Fig. 2. Here, we assumed that this semivariogram could be applied to interpolate WSN measured soil moisture in the period from June 10, 2012 to July 15, 2012.

A spherical model was used in this study as the semivariogram model, which is extremely important for structural analysis and spatial interpolation (Burrought and McDonnel, 1998).

$$\gamma(h) = \begin{cases} C_0 + C_1 \left[ 1.5 * \left( \frac{h}{a} \right) - 0.5 * \left( \frac{h}{a} \right)^3 \right], 0 \leq h \leq a \\ C_0 + C_1, \qquad\qquad\qquad\qquad\quad h > a \end{cases} \qquad (10)$$

where $\gamma(h)$ is the semivariance; $C_0$ represents a nugget, which is the minimum variability observed or the "noise" at a distance of zero; $C_1$ is the structural variance, $C_0 + C_1$ represents the sill variance; and $a$ is the range that signifies the correlation length

in geostatistics.

The Kriging method requires the second-order stationarity for geostatistical inference and assumes it to be isotropic. As our study area was in the central part of the oasis, and no significant soil wetness spatial trend could be found, the soil moisture observations in the study area were assumed to meet the above requirements. Because the area of the oasis is limited, i.e., the closest desert is about 4 km away from the center of study area, the semivariogram statistics beyond 4 km may be affected by

the presence of desert. Therefore, the range of the spherical model was set as 2500 m. Figure 3 shows the fitting curve of the semivariogram obtained using the spherical model.

### 3.2   Extending the Kriging method to incorporate remote sensing information

To reflect more details of the spatial distribution pattern of soil moisture, we propose a new algorithm that incorporates remote sensing variables, i.e., NDVI and albedo, into the basic Kriging method. The traditional interpolation space is the spatial space depicted by x and y coordinates. The new algorithm extends the interpolation space to the combined spatial and spectral space,

in which NDVI and albedo are treated as coordinates, just like $x$ and y. The distance in the combined space is characterized by the spatial distance and the spectral distance, as follows:

$$h = \sqrt{\Delta x^2 + \Delta y^2} \qquad\qquad (11)$$

$$s = \sqrt{\left( \frac{\Delta \text{NDVI}}{\sigma_{\text{DNVI}}} \right)^2 + \left( \frac{\Delta \text{albedo}}{\sigma_{\text{albedo}}} \right)^2} \qquad (12)$$

where $h$ is the spatial distance, $\Delta x$ and $\Delta y$ are the coordinate differences between two sampled points, $s$ represents the spectral distance, $\Delta$NDVI and $\Delta$albedo are the differences of NDVI and albedo values between two sampled points, and $\sigma_{\text{DNVI}}$ and $\sigma_{\text{albedo}}$ are two normalization factors (in this study, we simply set their values as 0.1, 0.1). The NDVI and albedo of each 30 m pixel in the study area were derived from the remote sensing images, after geometric and atmosphere correction. Correspondingly, the semivariogram model was extended to combined space, as signified in the equations below:



$$\gamma_1(h) = \begin{cases} C_1 * \left[ 1.5 \left( \frac{h}{a_1} \right) - 0.5 (\frac{h}{a_1})^3 \right], & 0 \le h \le a_1 \quad (13) \\ C_1, & h > a_1 \end{cases}$$

$$\gamma_2(s) = \begin{cases} C_2 * \left[ 1.5 \left( \frac{s}{a_2} \right) - 0.5 (\frac{s}{a_2})^3 \right], & 0 \le s \le a_2 \quad (14) \\ C_2, & s > a_2 \end{cases}$$

$$\gamma(h, s) = \gamma_1(h) + \gamma_2(s) + C_0 \qquad (15)$$

where $\gamma_1$ and $\gamma_2$ are the semivariogram values with respect to $h$ and $s$, $\gamma$ is the overall semivariogram, and $a_1$ and $a_s$ are the lag distances of the spatial and spectral variables.

We also used the soil moisture map to derive the semivariance statistics. The semivariance, as a function of $h$ and $s$, is shown in Fig. 4.

Here, the X-axis is spatial distance, the Y-axis is spectral distance, and the color in each grid represents the average semivariance value of the soil moisture. When h > 4000 and s > 4, the sampled data quantity is not sufficient to satisfy statistical significance. Therefore, we divided $h$ into six intervals ranging from 0 to 4000 m, and $s$ into six intervals ranging from zero to four; the semivariance in Fig. 4 is the average value in these intervals.

Using the semivariagram model as in the above Eq. (13), Eq. (14), and Eq. (15), the fitting semivariance diagram can be obtained, as shown in Fig. 5. The $a_1$ of spatial distance was preset as 2500 m, and the $a_2$ of spectral distance was preset as 3.5. Then, the fitted parameters $C_0$, $C_1$, and $C_2$ were 6.4468, 2.8762, and 2.9972, respectively.

## 4  Results and discussion

To evaluate the proposed Extended Kriging method, we first visually inspected the interpolation results and compared them with the results obtained using the tradition Kriging method. Then, we examined the precipitation and irrigation data to ascertain whether the temporal variation of the interpolated soil moisture was consistent with the water input. Subsequently, a quantitative uncertainty analysis was conducted with both Kriging's internal uncertainty estimator $\sigma_k$, and the leave-one-out cross-validation method. Finally, we determined that if a high-resolution remote sensing image of surface temperature is available, the interpolation accuracy could be significantly improved with this additional spectral information.

### 4.1  Spatio-temporal trend analysis

A spatial map of the interpolated soil moisture can directly reflect the capability of the interpolation methods to obtain the parameter's variation pattern. Compared with the original Kriging method, the advantages of the new Extended Kriging method can be visually observed in the Spatio-temporal pattern of the interpolation results. We used the semivariogram obtained from Sect. 3.2, soil moisture data of WSN at a depth of 10 cm on July 10, 2012, and the remote sensing data (NDVI and albedo) on July 13, 2012, to estimate the soil moisture distribution of the experimental area. The interpolation results





obtained for these two methods are shown in Fig. 6, where (a) is the result of the original Kriging method and (b) is the result of the Extended Kriging method. Both result maps are masked to exclude residential areas.

It is clear that the result for the Extended Kriging method shows more details of the field soil moisture distribution than that for the traditional Kriging method. The interpolation result for the traditional Kriging method can only show a smooth and continuously changing trend surface (Fig. 6a). In contrast, from the Extended Kriging interpolation result, it is clear that the soil moisture is not smooth and continuously changing. Figure 6b reveals a few rough veins in the ground surface and subtle changes in the field soil moisture.

In addition to measuring data from multiple sites simultaneously, WSN is also able to obtain real-time and continuous observations. This enabled us to estimate the continuous soil moisture distribution of the study area on both spatial and temporal domains. We interpolated the soil moisture distribution for every day from June 10, 2012 to July 15, 2012. Consequently, we obtained six interpolation maps, with seven-day intervals, as shown in Fig. 7. As can be seen, the spatial distribution of the soil moisture content changed significantly over time, indicating the necessity for dynamic monitoring of soil moisture. The drought areas and their borders can be clearly identified in the interpolated maps, which is very useful information for agricultural management.

## 4.2    Correlation with precipitation/irrigation data

To verify the authenticity of the time series of the interpolated soil moisture map, we chose several irrigation fields in the study area and compared the estimated results on temporal series from June 10, 2012 to July 15, 2012 with the precipitation and irrigation data in the same temporal series. Nine typical irrigation fields were chosen, with their names and locations as shown in Fig. 8: Jingcheng-6 (JC-6), Shiqiao-1 (SQ-1), Shiqiao-2 (SQ-2), Shang Touzha-1 (STZ-1), Wuxing-2 (WX-2), Wuxing-4 (WX-4), Wuxing-5 (WX-5), Xiaoman-5 (XM-5), and Xiaoman-9 (XM-9). We averaged the soil moisture values of all 30 m pixels in the irrigation field to refer to the soil moisture value of the field. The precipitation data were acquired from the super automatic meteorological station (Liu et al., 2011; Li et al., 2015; Liu et al., 2016) which is near the center of the study area. The irrigation data used were provided by the Cold and Arid Regions Sciences Data Center at Lanzhou (http://westdc.westgis.ac.cn). Although the records were not complete (some omissions exist), the available parts were still sufficient to meet the demand in this study. The comparison results are shown in Fig. 9. The X-axis is the experimental time, the Y-axis on left is the precipitation amount, and the Y-axis on the right is the estimated soil moisture content. The red point represents the irrigation status on that day. As shown, the precipitation records indicate that there was rain on June 17, June 26, June 27, July 3, July 6, July 8, and July 15. Furthermore, the irrigation dates are different for different fields. It is clear that the soil moisture value increases in response to the precipitation or the irrigation, and then gradually decreases as the fields dry out.



### 4.3 Uncertainty analysis

The estimation results for this Extended Kriging method not only show obvious advantages under the visual inspection, but also have a higher accuracy than that for the original Kriging method under quantitative analysis. The leave-one-out cross-validation method was adopted to estimate the interpolation accuracy. In the leave-one-out cross-validation method, every sample is served as a testing sample; assuming that the number of samples is N, then there are N-1 testing samples each time and, finally, N testing results. The Root Mean Square Error (RMSE) of these N testing results is used to evaluate the interpolation accuracy. Figure 10 is the time series of the average RMSE of all available WSN nodes, derived from the leave-one-out method. As is seen from Fig. 11, the RMSE value of the Extended Kriging method is also smaller than that of the original Kriging method, which indicates that the proposed new interpolation method is quantitatively more accurate.

The last paragraph in Sect. 3.1 introduced $\sigma_k$ as an uncertainty assessment of the Kriging algorithm itself, which indicates the intrinsic uncertainty brought by the algorithm and the influence of the environment. We calculated the $\sigma_k$ and RMSE of these two interpolation methods from June 10 to July 15. Figure 11 is the time series of the average $\sigma_k$ of all the pixels in the study area. As can be seen in Fig. 11, the $\sigma_k$ of the Extended Kriging method is always smaller than that of the original Kriging method, which means that, theoretically, introducing the spectral information can reduce the interpolation uncertainty.

Although both the RMSE and $\sigma_k$ values for the Extended Kriging method are smaller than those for the original Kriging method are, the influence of external interference on interpolation accuracy is still evident. The several obvious high RMSE values in Fig. 10 are close to the dates of precipitation. It is possible that the precipitation brings extra spatial and temporal variations to soil moisture. However, there is a second explanation: because the remote sensing image was not available around the precipitation dates, we had to use an image that was acquired on an earlier date. For example, the interpolation result for June 28 is based on the image acquired on June 19; thus, this large uncertainty resulted from temporal mismatch between remote sensing data and ground data.

### 4.4 Effect of adding temperature information to the interpolation method

Land surface temperature (LST) is an important index connected with soil moisture, usually appearing as negatively correlated with soil moisture. Therefore, it is a potential spectral index that can support interpolation of soil moisture. However, high-resolution temperature remote sensing data are not as widely available as those of NDVI and albedo are. Fortunately, an LST map at 2.5 m resolution is available for July 10, 2012, as hyperspectral thermal infrared airborne images were acquired by the TASI instrument on that date. Therefore, we added the LST as the third spectral index, together with NDVI and albedo, in the interpolation of this date, and compared the result with that obtained prior. Equation (12) is revised as Eq. (16). Estimation maps are shown in Fig. 12, where (a) is the result with the indexes of NDVI and albedo and (b) is the result with the indexes of NDVI, albedo, and LST. The comparison of uncertainty is shown in Table 1.

$$s = \sqrt{(\Delta\text{NDVI}/\sigma_{\text{DNVI}})^2 + (\Delta\text{albedo}/\sigma_{\text{albedo}})^2 + (\Delta\text{LST}/\sigma_{\text{LST}})^2} \quad (16)$$





Here, ΔLST represents the difference value of temperature between two sampled points from the LST map, which is aggregated to 30 m resolution, and $\sigma_{LST}$ is the normalization factors, the value of which is one.

The result indicates that introducing a new spectral temperature index can further improve the accuracy of soil moisture content: the value of $\sigma_k$ is reduced from 9.2070 to 8.0043 and the value of RMSE from 1.6406 to 1.3958. Hence, if high-resolution LST data are available in a long time series, the future of soil moisture interpolation can incorporate the LST information to boost accuracy.

## 5  Conclusion

With the rapid development of ground-based Earth observing techniques such as wireless sensor network, we are now able to monitor environmental parameters in real time, continuously, and with multiple sample points. However, interpolation is still needed to extend the point measurement to spatial distribution of the corresponding parameter in an area. As satellite remote sensing is an efficient way of acquiring area Earth observing data, it is desirable to combine information from remote sensing and from ground-based observation networks.

The Extended Kriging method proposed in this study introduced the remote sensing image spectral information into the traditional interpolation method. NDVI and albedo are the spectral variables used in the algorithm. These spectral variables are treated in the same manner as the spatial variables, i.e., $x$ and $y$. Therefore, the interpolation is fundamentally the same Kriging algorithm, but operating on the combined space of spatial dimension and spectral dimension. The semivariogram model is also extended to the combined space. A remote sensing derived soil moisture map is used in this paper to fit the semivariogram model. However, this soil moisture map can be replaced by other sources of samples as long as the dataset is sufficiently large to derive robust statistics about the semivariance.

The proposed algorithm was applied to the soil moisture dataset acquire by the soil moisture sensors network (WATERNET) in the oasis agricultural areas, which is the forci experimental area of the HiWATER campaign. As the WATERNET provides continuous near-surface soil moisture measurement over 48 scattered points, the interpolation results are daily soil moisture maps from June 10, 2012 to July 15, 2012, covering an area approximately 4.5 km × 5.0 km in size. Visual inspections indicate that the interpolation result from the proposed Extended Kriging algorithm presents much more spatial details than that of the traditional Kriging algorithm. The field-average soil moisture of several irrigation fields for long time series are associated with the precipitation data and irrigation data, and the temporal variation of soil moisture can be well explained by these water inputs. The quantitative uncertainty analysis with both the leave-one-out method and $\sigma_k$ indicate that the Extended Kriging algorithm, which operates in the spectral and spatial combined space, produces more accurate interpolation results than that of the traditional Kriging algorithm, which operates only in the spatial domain. Currently, NDVI and albedo are recommended as the spectral variables to aid interpolation because they can be easily derived from most high-resolution satellite images. However, we demonstrated in the discussion that more relevant spectral variables, such as land



surface temperature, could be incorporated into this algorithm to improve its performance. However, how to choose the informative spectral variables remains an open topic for this algorithm.

There are other methods that can combine information from ground measurement and information from remote sensing, e.g., the Co-Kriging method and the data assimilation method (Gao et al., 2014). However, we prefer not to compare them

with the proposed algorithm in terms of accuracy for the following two reasons: in the first place, the Extended Kriging algorithm is much simpler than the Co-Kriging method and the data assimilation method. Second, the Extended Kriging algorithm utilizes easily available variables such as NDVI and albedo, which are difficult to use in the Co-Kriging method and the data assimilation method because NDVI and albedo do not have a clear physical connection with soil moisture. The Extended Kriging algorithm has potential, as well as space for improvement. For example, the normalization factors for

spectral variables could be refined; and the semivariogram model in the combined space is too simple. Nevertheless, we prefer to present the simplest form of the algorithm to the reader, and leave these improvements to future researchers.

There are other aspects that we could not delve into deeper in this short paper. One of them is the quality, or accuracy, of WSN-measured soil moisture. As we know, it is technically difficult to install Hydro Probe II (HP-II) sensors exactly at 4 cm and 10 cm below the surface; and the WSN sensors are more or less infected by its internal voltage and temperature; and the

15 sensors can go to saturation in extremely low or high soil moisture values (Gao et al., 2014; Zhang et al., 2015). It is also true that the remote sensing data may suffer from inaccurate calibration and atmospheric correction. In addition to the flow of data, the scale mismatch between the footprint of WSN nodes and 30 m resolution remote sensing pixels should be considered. Nevertheless, currently we do not have solid data to support a study on these aspects.

Another related interesting topic is the desired density of the WSN nodes, and how to optimize the location of the nodes.

Fortunately, a similar topic has been addressed by other researchers (Wu et al., 2016), although their research area and target parameter are different. In this paper, we simply used all the valid WATERNET data.

The proposed algorithm is a development of the classical Kriging method. Although it is proposed in this paper to interpolate the soil moisture data, it is potentially applicable to other environmental parameters. As new observation technologies are being applied wider, increasingly more high-quality measurement data, at multiple sites in a small area, will

become available to the public, and the potential of the classical Kriging can be further explored.

**Acknowledgments**

This study was supported by the National Natural Science Foundation of China (No. 41331171 and No. 41476161), Special Fund from Chinese Academy of Sciences (No. KZZD-EW-TZ-18), and the Open Fund of the State Laboratory of Remote Sensing Science (No. OFSLRSS201626). The ground data were provided by the Cold and Arid Regions Sciences Data Center

at Lanzhou. We would also like to thank the researchers in the Institute of Remote Sensing and Digital Earth (RADI), Chinese Academy of Sciences for their valuable discussions as well as their data processing support.





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

Figures:





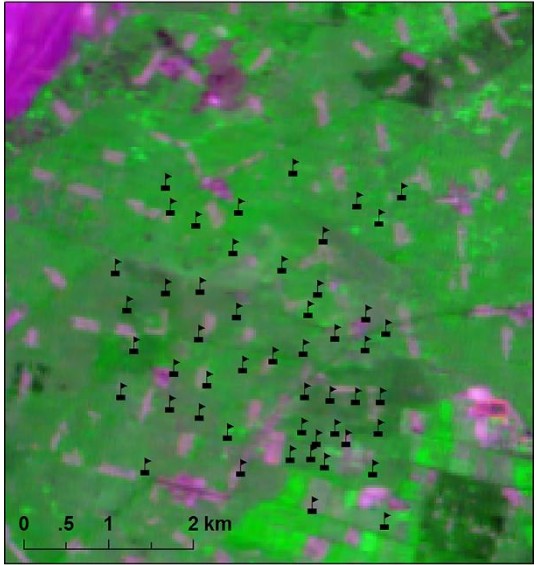

Figure 1 Study area in the middle reaches of the Heihe River Basin (the black flags indicate the location of WATERNET nodes)

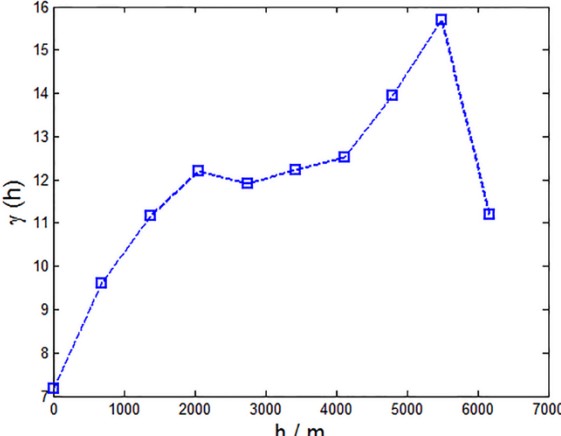

Figure 2 Semivariance of sampled soil moisture data





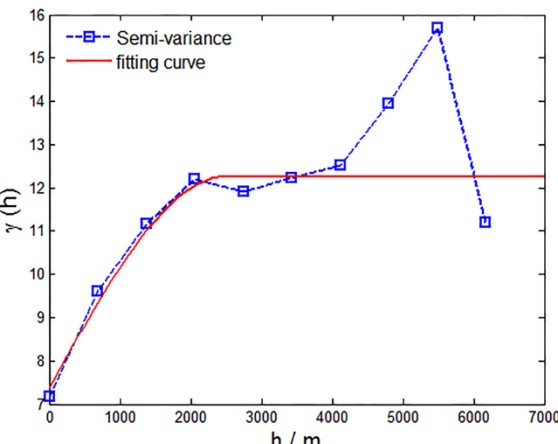

Figure 3 Fitting curve of the soil moisture semivariogram

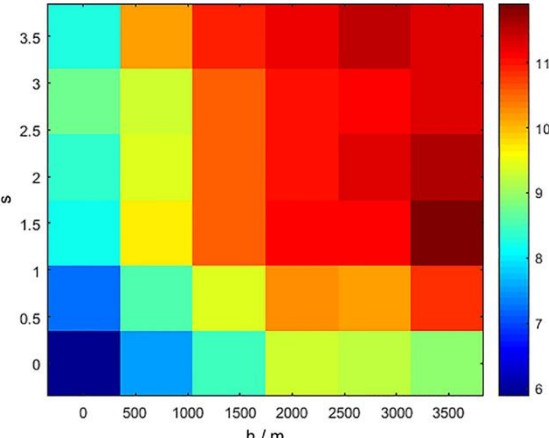

Figure 4 Semivariance of sampled soil moisture data with respect to spatial and spectral distance

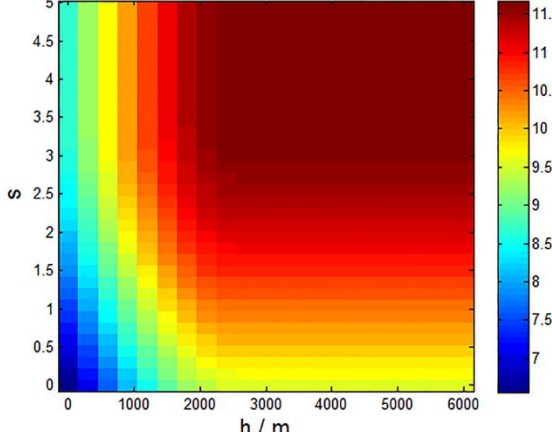

Figure 5 Fitting result for soil moisture semivariogram in the spatial and spectral dimensions





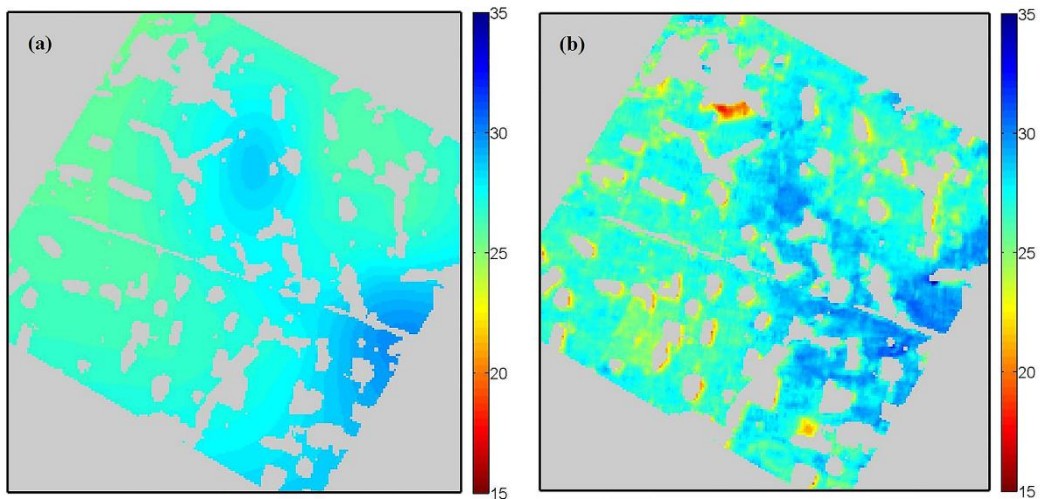

Figure 6 Interpolated soil moisture at depths of 10 cm on July 10, 2012:

(a) Interpolation result for traditional Ordinary Kriging method; (b) Interpolation result for Extended Kriging method

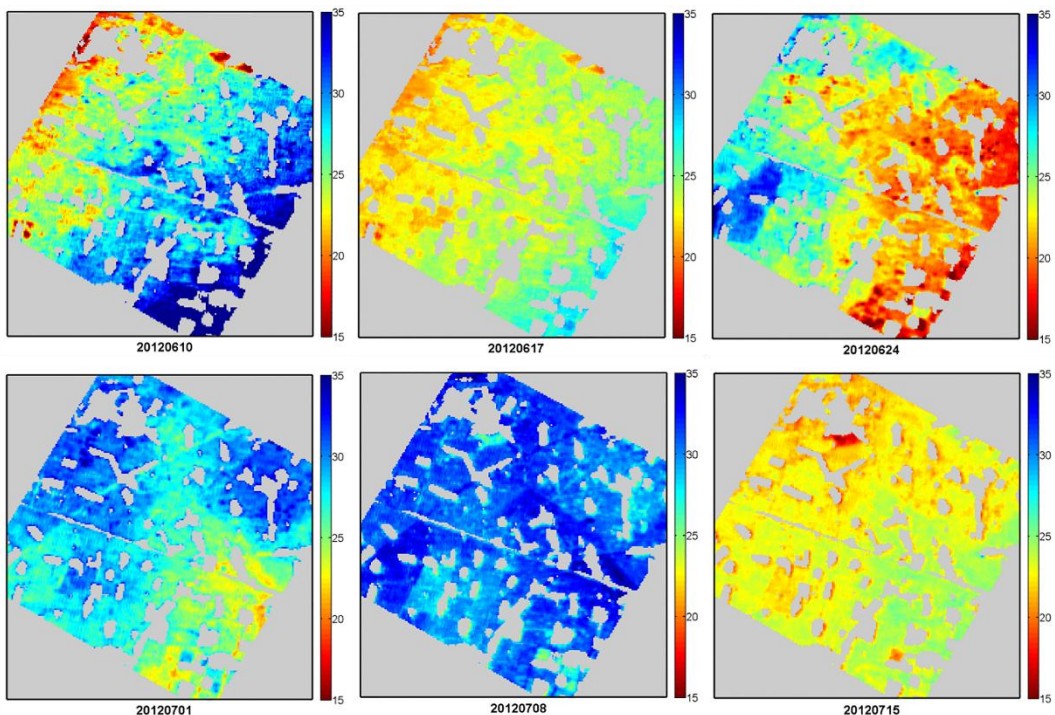

5 Figure 7 Spatio-temporal distribution of soil moisture in the study area





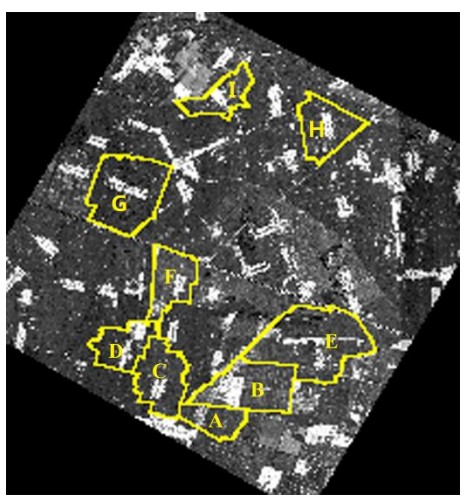

Figure 8 Location of the selected irrigation fields

A is WX-1, B is WX-4, C is XM-5, D is XM-9, E is WX-5, F is XM-3, G is SQ-2, H is KN-8, and I is SYZ-1

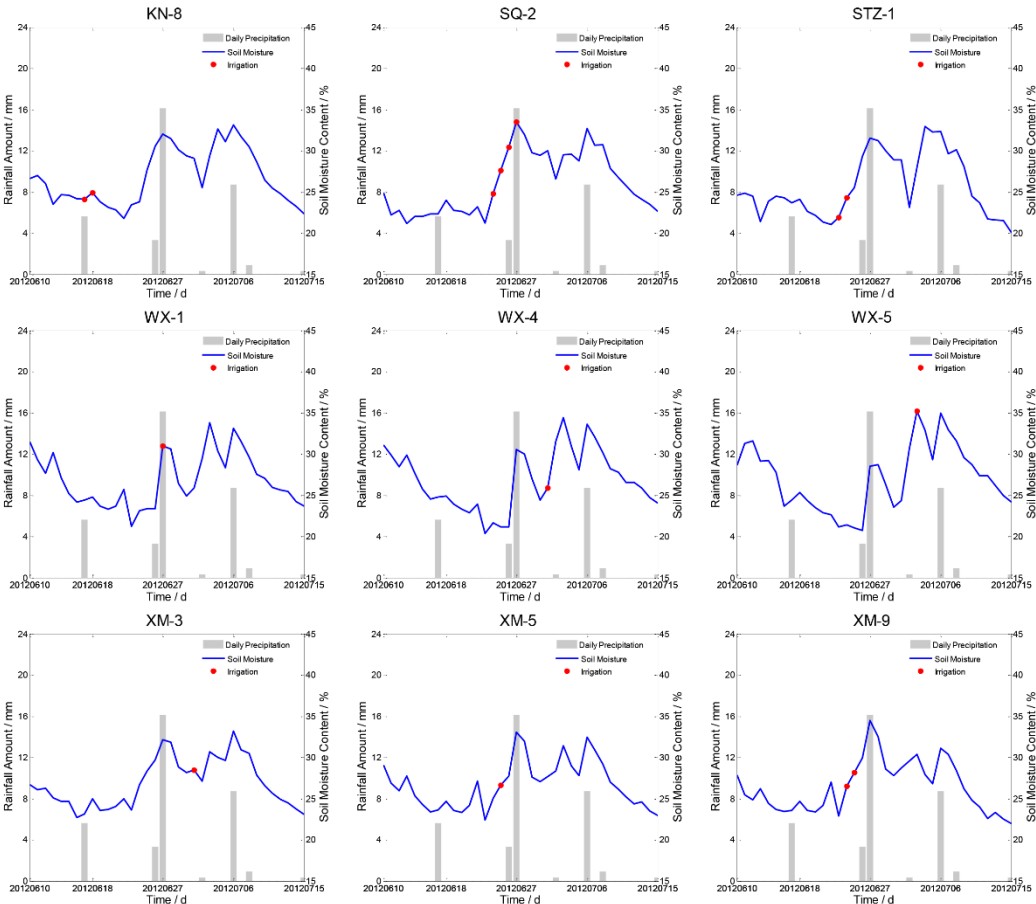

5          Figure 9 Comparison of soil moisture changing trend with precipitation and irrigation data of the chosen irrigation fields





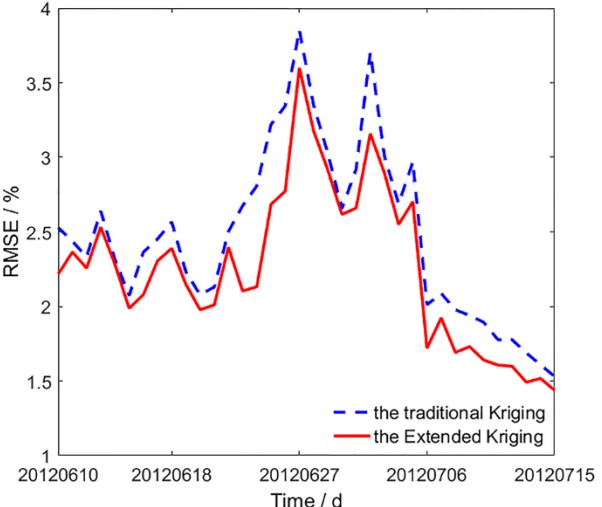

Figure 10 Comparison of the RMSE indicator

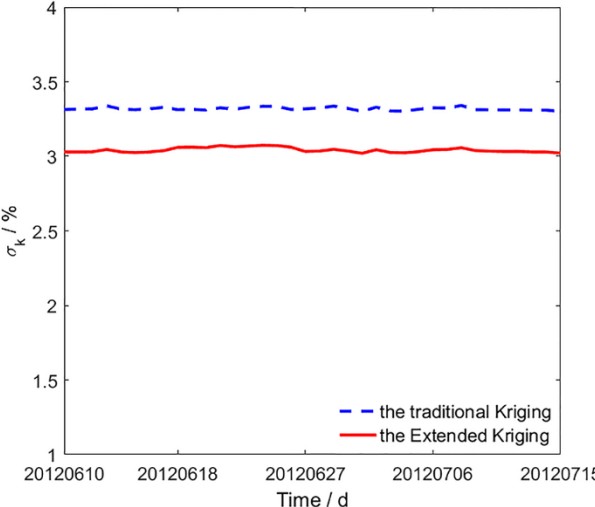

Figure 11 Comparison of the $\sigma_k$ indicator





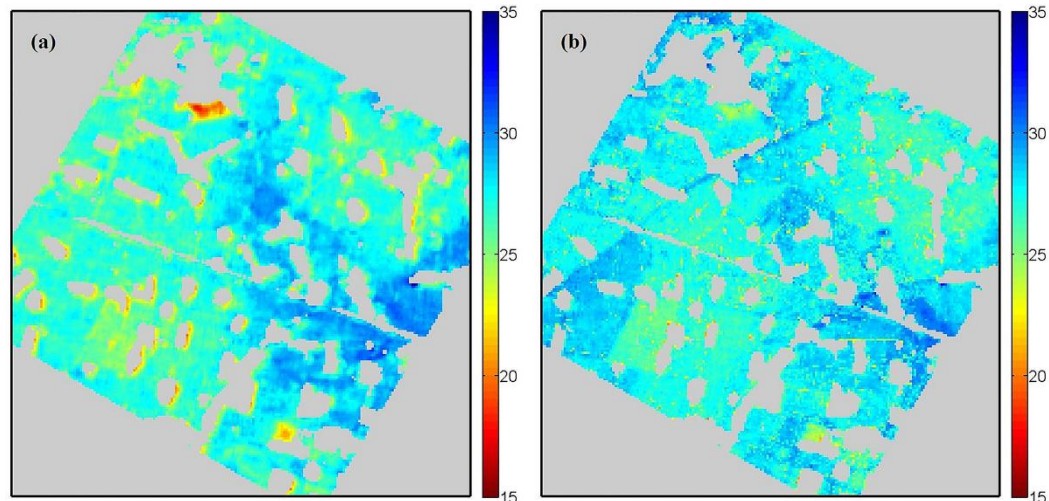

Figure 12 Soil moisture interpolation results at a depth of 10 cm on July 10, 2012: (a) Using the Extended Kriging method with the spectral indices of NDVI and albedo; (b) using the Extended Kriging method with the spectral indices of NDVI, albedo, and LST

5   Table:

Table 1 Comparison of the interpolation results for the traditional Kriging method and the Extended Kriging method

|  | Traditional Kriging | Extended Kriging with NDVI and albedo | Extended Kriging with NDVI, albedo and LST |
|---|---|---|---|
| $\sigma_k$ | 10.9725 | 9.2070 | 8.0043 |
| RMSE | 1.8937 | 1.6406 | 1.3958 |