# Peer review of "An Extended Kriging method to interpolate soil moisture data measured by wireless sensor network"

_Hydrology and Earth System Sciences, 2016_

## Referee Comment (RC1) · Anonymous Referee #1 · 10 Oct 2016

General comments:

This interesting study published in HESSD proposes a technique for the spatial interpolation of soil moisture measurements obtained by wireless sensor networks (WSN). Remote sensing data of NDVI and albedo are used as an additional information in the novel approach referred as Extended Kriging. The acronym NDVI is not clearly defined in the text and it is assumed that it refers to the vegetation index abbreviated by VI in line 17 of page 1. The interpolation technique is based on transferring the standard spatial assumptions of Ordinary Kriging to a combination of spatial distance and additional information related assumptions.

The results are presented in a good and mostly comprehensible form and the structure of the paper is reasonable. However, the manuscript contains some spelling and

grammar errors and the formulations could be more concise at some points. The figure captions could generally contain more information. Furthermore, it is not entirely clear how the interpolation techniques were applied. In total, there are several points that need further explanation and require additional work. Due to this, I recommend the paper to be returned for major revision.

Specific comments:

(1) Data pre-processing is often crucial for the interpolation performance of geostatistics. The information given in Sec. 2.2 is not sufficient. It should be explained how the exclusion of abnormal WSN data was performed.

(2) It is not clear why vegetation index in combination with albedo was selected as additional information. This needs to be discussed more. A short correlation analysis for several variables might help to justify this choice.

(3) The equations of Ordinary Kriging (Eq. 1, 2, 3 and 4) are not explained entirely. For many readers it might be clear that n refers to the number of adjacent measurements taken into account for the spatial estimation, nevertheless it should be mentioned somewhere in the text. The actual number of points considered for the interpolations using Ordinary Kriging and Extended Kriging should be mentioned as well in the methodology.

(4) It is not entirely clear how the variogram fitting was conducted and whether an automatic or manual approach was used for this. What exactly is shown in Figs. 2, 3, 4 and 5? Are these average experimental semivariograms or the experimental semivariograms of a specific time step? Figure 2 can be omitted.

(5) It is not sufficiently described how the interpolation was performed. The 5 min WSN measurements were aggregated to daily estimates of soil moisture, but only five clear-sky satellite images were available for specific days. Did you apply the ordinary kriging and Extended Kriging interpolation only for these five time steps? The performance

curves (Figs. 10 and 11) show more than five sampling points. Theoretically, all days of the investigation period need to be interpolated. Is it possible to apply this method for days without satellite information, for instance by using averaged spectral variables? This might be particularly interesting for the implementation of temperature data.

(6) I suppose that the correlation discussed in section 4.2 refers to the correlation of soil moisture maps interpolated by Extended Kriging with precipitation and irrigation data. Is it possible to show also the correlation using Ordinary Kriging? It might be interesting to see whether the implementation of satellite information can improve it.

(7) I strongly recommend comparing the performance of Extended Kriging with the performance of a standard multivariate geostatistical technique, for instance Cokriging or Kriging with External Drift. It is true that Extended Kriging is somewhat simpler. Nevertheless, it would be useful to achieve a better indication of interpolation performance. The second objection stated in the lines 6 to 8 on page 12 is not valid. Multivariate geostatistics is often applied to data without a direct physical relation.

Minor technical corrections:

(1) What is the reason for the tilted perspective or the masking of the borders in the maps of Figs. 6 and 7? Simple two-dimensional plots might be a better solution. I recommend preparing Fig. 1 in a consistent way, i.e. use the same masking. What is the background colour shading in Fig. 1?

(2) The term uncertainty analysis refers usually only to evaluations regarding the kriging standard deviation. I suggest renaming section 4.3 to cross validation and uncertainty analysis.

(3) Page 3, line 30 and other occurences: The correct spelling is cokriging.

(4) Page 7, lines 5-6: Why is the spherical model in particular important for structural and spatial interpolation? I recommend removing this clause.

---

## Author Comment (AC1) · 16 Oct 2016

Thanks for your review. The comments and suggestions are very helpful. Here, I give some quick explanations to several short questions that I didn't elaborate clearly in the paper. For the others, I will reply as soon as possible after further analysis.

1. specific comment (4):

It is not entirely clear how the variogram fitting was conducted and whether an automatic or manual approach was used for this. What exactly is shown in Figs. 2, 3, 4 and 5? Are these average experimental semivariograms or the experimental semivariograms of a specific time step? Figure 2 can be omitted.

Reply:

[Figure]

As the number of WSN nodes installed in study area are limited, the node amount is insufficient to gather robust statistics in the process of semivariogram fitting. Thus, we used a soil moisture map of the study area to obtain sampling data. The soil moisture map was derived from the airborne hyperspectral datasets of CASI/TASI, acquired on July 10, 2012.

In the calculation, we sampled 9000 points from the soil moisture map as the sampling data for semivariogram fitting. To reduce the error, the spatial/spectral distance was divided into ten bins. We averaged the semivariance values of each bin as the final semivariance value of each distance. As the amount of the sampling data was insufficient when the spatial/ spectral distance was increased up to a certain extent, its average semivariance value was not stable. Therefore, these invalid data were removed before semivariogram fitting. For example, we only used the data when h<=4000 m in the fitting process in Fig. 3. Figure 4 shows the valid average semivariance data after manual exclusion. The color of each grid in Figs. 3 and 4 presents its average semivariance value.

In the process of semivariogram fitting, the nugget value and sagitta value were calculated by automatic approach, and the range distance was set by manual approach.

As the sampling data were acquired from the soil moisture map mentioned above, the experimental semivariograms correspond to the time of the soil moisture map. In this manuscript, we assume that the derived semivariograms can be applied to all the dates of interpolation.

2. specific comment (5):

It is not sufficiently described how the interpolation was performed. The 5 min WSN measurements were aggregated to daily estimates of soil moisture, but only five clearsky satellite images were available for specific days. Did you apply the ordinary kriging and Extended Kriging interpolation only for these five time steps? The performance curves (Figs. 10 and 11) show more than five sampling points. Theoretically,

all days of the investigation period need to be interpolated. Is it possible to apply this method for days without satellite information, for instance by using averaged spectral variables? This might be particularly interesting for the implementation of temperature data.

Reply:

As only five clear-sky satellite images were available for specific days, we supposed that the satellite images would not change much in a few days. In this way, for days without satellite images, we applied the available satellite image with nearest date to their interpolation. Thus, we could obtain the interpolation results in continuous time series.

3. Minor technical correction (1):

What is the reason for the tilted perspective or the masking of the borders in the maps of Figs. 6 and 7? Simple two-dimensional plots might be a better solution. I recommend preparing Fig. 1 in a consistent way, i.e. use the same masking. What is the background color shading in Fig. 1?

Reply:

The reasons for the masking of the borders in maps of Figs. 6 and 7 are as follows: (1) WSN nodes in the study area distributed within the borders of Fig. 6; (2) there are no land surface temperature data outside the borders of figure 6. Thus, we finally used the same masking for all the interpolation results.

The image shown in Fig. 1 was obtained from HJ satellite, combined by band 3(RED), band 4(NIR) and band2(GREEN).

---

## Referee Comment (RC2) · Anonymous Referee #2 · 4 Nov 2016

This paper introduces an interesting alternative model for prediction of a spatial variable by kriging. However, it does not reflect adequately on the motivation of the model, or explore its implications. There is therefore no real justification for the selection of the model, which strikes me as very implausible. Furthermore, the model is not implemented correctly. I expand on these statements below.

First, there is no reason not to extend the geostatistical model from 1, 2 or 3 spatial dimensions to a space of higher dimensions. This is done, afterall, in space-time geostatistics. However, if one is to treat some covariate as defining an additional dimension then, if one is assuming intrinsic stationarity in the new space, as one must for this extension of ordinary kriging, then this implies that there is no systematic relationship between the target variable and the covariate. In terms of the definition of the intrinsic hypothesis of stationarity E[Z(s)-Z(s+h)]=0 where Z(s) defines our random variable ex-

[Figure]

pressed as a random function of the covariate. In words, one is envisaging a situation in which a plot of the observations z against the corresponding values of the covariate s would not show any systematic trend but rather a random fluctuation, exhibiting some degree of correlation. This seems a rather implausible model to me. However it might be worth considering. Despite what the authors say, however, it is not a model one might use in a situation where cokriging or kriging with external drift were also candidate approaches, since in the case of kriging we assume a linear relationship between z and s, and in KED we assume a linear relationship or a relationship linear in the parameters of some simple fixed effects structure such as a polynomial or (non unduly complex) spline basis. This proposed approach and cokriging/KED would be fundamentally incompatible. Given this, one would expect the authors so start by showing that cokriging or KED are not suitable for these data by plots and exploratory statistics that show (for cokriging) that a linear model of coregionalization is not plausible or (for KED) that no reasonable fixed effects structure looks reasonable.

Let us assume that the authors do show a sound motivation for applying their model. They must then estimate it appropriately. I think there is a difficulty here. The authors should read the literature on space-time geostatistics to get a better understanding of this. One problem is that, unlike the space-time case, one cannot compute estimates of two marginal variograms (the spatial variogram with the time (or here s) lag =0, and vice versa). In this paper the authors cannot compute a marginal variogram in s space, because it is not possible to find observations with lag 0 in space but some non-zero lag in s. The variogram shown presumably uses lag bins, but the lack of the marginal variogram is a problem because there is likely to be nugget variation in both dimensions and you cannot resolve their contribution to the joint space-s variogram. There is a second problem. The authors propose a simple model for the space-s variogram which is the sum of a spatial and an s-dimension variogram (Equation 15), but such a model is not in general valid (i.e. it does not define a non-negative definite covariance structure in the overall space). The authors should look at papers such as the one by De Cesare, L., Myers, D.E., Posa, D., 2001.[ Estimating and modelling

space–time correlation structures. Stat. Probab. Lett. 51, 9–14.] for an account of this and of some valid models.

I have a further problem. The authors are using the variogram from a remote sensor for kriging from in-situ sensors. Even if they could be confident that the two variables are measurements of the same underlying quantity the variogram of the remote sensor data is a regularization of the variogram of the networked sensor data onto a very different spatial support. The question of how the remote sensor data might be used to help with this problem is interesting, given the fact that you have some data on the desired support it might be possible to do this, but it requires an explicit change of support step, not just ignoring the difference.

---

## Author Comment (AC2) · 29 Nov 2016

General comments: This interesting study published in HESSD proposes a technique for the spatial interpolation of soil moisture measurements obtained by wireless sensor networks (WSN). Remote sensing data of NDVI and albedo are used as an additional information in the novel approach referred as Extended Kriging. The acronym NDVI is not clearly defined in the text and it is assumed that it refers to the vegetation index abbreviated by VI in line 17 of page 1. The interpolation technique is based on transferring the standard spatial assumptions of Ordinary Kriging to a combination of spatial distance and additional information related assumptions. The results are presented in a good and mostly comprehensible form and the structure of the paper is reasonable. However, the manuscript contains some spelling and grammar errors and the formulations could be more concise at some points. The figure captions could gener-

ally contain more information. Furthermore, it is not entirely clear how the interpolation techniques were applied. In total, there are several points that need further explanation and require additional work. Due to this, I recommend the paper to be returned for major revision.

ANS:

The anonymous Referee#1 gave us many constructive suggestions and comments. Following his suggestions, the authors have made major revision to the manuscript: besides the changes in the text, a new section (Section 4) is inserted into the manuscript to present the comparison results between Cokriging and the proposed Extended Kriging algorithm, as well as the analysis about the effect of number of observation points to be interpolated. After the revision, the motivation for proposing the new algorithm, instead of using Cokriging or Kriging with External Drift, is explained more clearly. We are very grateful for the referee's help.

We have posted a quick short response to Referee#1 before. But it is not enough. Now we will formally answer Referee#1's questions one by one.

Specific comments:

Specific comments (1): Data pre-processing is often crucial for the interpolation performance of geostatistics. The information given in Sec. 2.2 is not sufficient. It should be explained how the exclusion of abnormal WSN data was performed.

ANS:

Data pre-processing is absolutely necessary. A short description of data pre-processing is added into section 2.2. As we know, the WATERNET is an experimental WSN established for the campaign in summer 2012. Some of the sensor nodes are affected by sensor noise and abnormal conditions during wireless data transfer. Smoothing and noise reduction treatments are necessary. Besides, when the soil moisture is close to saturation, the soil moisture sensors cannot work properly and sometimes give

abnormal values. Thus, this part of data also need be removed. We first excluded the abnormal data by assigning the zero value, negative value and abnormally high value (soil moisture content > 50 percent) as invalid data NaN. Then, we averaged the valid data for the whole day, and used the average result as the final soil moisture value for each node.

Specific comments (2): It is not clear why vegetation index in combination with albedo was selected as additional information. This needs to be discussed more. A short correlation analysis for several variables might help to justify this choice.

ANS:

We are sorry that we neglected to explain the acronym NDVI (Normalized Difference Vegetation Index) in the old manuscript. We have added the definition of NDVI in Section 2.2 where it appeared for the first time.

Selecting NDVI and albedo as auxiliary information are out of the following two main considerations:

(a) NDVI and albedo are the spectral indexes which are fairly easy to be obtained from almost all high resolution remote sensing data sources. Although some other spectral indexes, such as TVID or NDWI, may have better ability to indicate soil moisture, they will require remote sensing data from thermal infrared or shortwave infrared channels, which usually are not available for high resolution satellite remote sensors. In order to develop a practical algorithm, it is imperative for us to choose the indexes from the practical data sources.

(b) NDVI and albedo represent most of the remote sensing information in the visible and near infrared spectral range. The raw remote sensing data are expressed as radiance (or reflectance) of multiple spectral channels. Because of the correlation between channels, the raw remote sensing data are redundant. In order to efficiently use the information in remote sensing data, principle component analysis (PCA) can be adopted

to extract the most informative component. Usually, the first and second principle component from most images will be the brightness and greenness, which approximately correspond to albedo and NDVI.

Although our reason of selecting NDVI and albedo as spectral variables is not based correlation analysis, it is still necessary to discuss the correlation of NDVI and albedo to soil moisture. So, we have added this correlation analysis into Section 3.2 of the revised manuscript. From the result of correlation analysis (see the revised manuscript), it is concluded that there is no significant LINEAR correlation between soil moisture and NDVI, or between soil moisture and albedo, the correlation coefficients are below 0.5. And this low correlation explains why the Cokriging algorithms cannot achieve satisfactory interpolation of soil moisture.

Specific comments (3): The equations of Ordinary Kriging (Eq. 1, 2, 3 and 4) are not explained entirely. For many readers it might be clear that n refers to the number of adjacent measurements taken into account for the spatial estimation, nevertheless it should be mentioned somewhere in the text. The actual number of points considered for the interpolations using Ordinary Kriging and Extended Kriging should be mentioned as well in the methodology.

ANS:

We have appended the explanation for the equations of ordinary Kriging in the revised manuscript.

As to the actual number of points considered for the interpolations, we plot the number of points for each day in Fig. 1 in this reply. The reason for the fluctuation of actually used numbers is the occasionally malfunctions of WSN nodes. These invalid observations are detected in the pre-processing and excluded in the interpolation. As the fluctuation is not prominent, we think it unnecessary to present this table in the paper. However, the range of the number of points is mentioned in the last paragraph of Section 5.1.

Specific comments (4): It is not entirely clear how the variogram fitting was conducted and whether an automatic or manual approach was used for this. What exactly is shown in Figs. 2, 3, 4 and 5? Are these average experimental semivariograms or the experimental semivariograms of a specific time step? Figure 2 can be omitted.

ANS:

As the number of WSN nodes installed in study area is limited, the node amount is insufficient to gather robust statistics in the process of semivariogram fitting. Thus, we used a soil moisture map of the study area to obtain sampling data. The soil moisture map was derived from the airborne hyperspectral datasets of CASI/TASI, acquired on July 10, 2012.

In the calculation, we sampled 9000 points from the soil moisture map as the sampling data for semivariogram fitting. To reduce the error, the spatial/spectral distance was divided into ten bins. We averaged the semivariance values of each bin as the final semivariance value of each distance. As the amount of the sampling data was insufficient when the spatial/ spectral distance was increased up to a certain extent, its average semivariance value was not stable. Therefore, these invalid data were removed before semivariogram fitting. For example, we only used the data when h<=4000 m in the fitting process in Fig. 3. Figure 4 shows the valid average semivariance data after manual exclusion. The color of each grid in Figs. 3 and 4 presents its average semivariance value.

In the process of semivariogram fitting, the nugget value and partial sill value were calculated by automatic approach, and the range distance was set by manual approach. As the sampling data were acquired from the soil moisture map mentioned above, the experimental semivariograms correspond to the time of the soil moisture map. In this manuscript, we assume that the derived semivariograms can be applied to all the dates of interpolation.

Figure 2 in the original manuscript has been removed in the revised manuscript.

[Figure]

Specific comments (5): It is not sufficiently described how the interpolation was per-formed. The 5 min WSN measurements were aggregated to daily estimates of soil moisture, but only five clearsky satellite images were available for specific days. Did you apply the ordinary kriging and Extended Kriging interpolation only for these five time steps? The performance (Figs. 10 and 11) show more than five sampling points. Theoretically, all days of the investigation period need to be interpolated. Is it possible to apply this method for days without satellite information, for instance by using aver-aged spectral variables? This might be particularly interesting for the implementation of temperature data.

ANS:

As only five clear-sky satellite images were available in the time range of interpolation, we assumed that the spectral variables derived from satellite images would not change much within a few days. This assumption is generally valid in the visible and near infrared remote sensing. In this way, for days without satellite images, we applied the available satellite image with nearest date to their interpolation. Thus, we could obtain the interpolation results in continuous time series.

In the case of temperature data, however, the problem is different. Because land sur-face temperature is a quick changing variable, it is not appropriate to the temperature data acquired. So, we think that temperature should not be applied to neighboring days as NDVI and albedo.

Specific comments (6): I suppose that the correlation discussed in section 4.2 refers to the correlation of soil moisture maps interpolated by Extended Kriging with precipitation and irrigation data. Is it possible to show also the correlation using Ordinary Kriging? It might be interesting to see whether the implementation of satellite information can improve it.

ANS:

The original Fig.9 has been revised as Fig.12 in the new manuscript. The interpolation results of ordinary Kriging are added into the figure. As is shown from the curves, the difference of these two methods is not prominent. This can be explained from two aspects: 1) The advantage of the Extended Kriging is to present more details of soil moisture spatial distribution. But averaging the interpolation result to the scale of field blurs the spatial details. 2) The main feature of the Extended Kriging is that it uses remote sensing image to aid spatial interpolation. But Fig.12 demonstrates the temporal variation of soil moisture and the information of temporal variation comes from the WSN data instead of the remote sensing image. So, Fig.12 actually is not suitable to demonstrate the main feature of the Extended Kriging.

The main purpose of the analysis in Sec. 4.2 is to prove that, by inheriting the temporal continuity of WSN observations, the result of Extended Kriging can reflect the changes of soil moisture on time series. So, the soil moisture derived from this approach is continuous both in spatial and temporal dimensions.

Specific comments (7): I strongly recommend comparing the performance of Extended Kriging with the performance of a standard multivariate geostatistical technique, for instance Cokriging or Kriging with External Drift. It is true that Extended Kriging is somewhat simpler. Nevertheless, it would be useful to achieve a better indication of interpolation performance. The second objection stated in the lines 6 to 8 on page 12 is not valid. Multivariate geostatistics is often applied to data without a direct physical relation.

ANS:

One of the reasons to design the Extended Kriging algorithm is that it is simpler to operate than other multivariate geostatistical techniques. So, Extended Kriging can be applied in cases where the pre-conditions of Cokriging or Kriging with External Drift are not satisfied. For example, when we first tried to interpolate the soil moisture measurements from WSN with the Cokriging software package in R language, the interpolation
result did not show enough spatial details, possibly because that the correlation of soil moisture with the spectral indexes are weak. Then, we tried the Extended Kriging algorithm, and the result looks better. These comparison results have been added into the revised manuscript as Fig.8 and Table 1 in Section 4.

In this analysis, we use the remote sensing retrieved soil moisture map to extract sampling points and validation points. Firstly, we sampled 9000 points from the soil moisture map to calculate the semivariogram of Cokriging method. Then we used different numbers of points, ranging from 300 points to 30 points, to interpolate soil moisture. The numbers of sampling points and RMSEs of interpolation results are shown in Table 1 of the revised manuscript. All the points of the soil moisture inversion map were used as validation points to calculate RMSE for each interpolation. As the locations of sampling points will influence the interpolation result and accuracy, we sampled randomly and repeated enough times to decrease the disturbing of point locations, and calculate the average RMSE value. From Table 1, we can see that the interpolation uncertainty (indicated both by $\sigma\_{\kappa}$ and RMSE) decreases while number of sampling points increases, and the estimator $\sigma\_{\kappa}$ can reflect the variation trend of the actual RMSE. We also find out that the RMSE of Extended Kriging and Cokriging are close, except that Cokriging performs a little better when the number of sample points is less than 50.

However, when we made a visual inspection to the interpolated soil moisture map (Fig.8 of the revised manuscript), it looks like that the interpolation of Extended Kriging can present more detailed information for the spatial distribution of soil moisture than that of Cokriging. These test results indicate that in cases where the sophisticated multivariate geostatistical techniques are not applicable, it may be worthy to try the Extended Krigning algorithm proposed in this manuscript.

The method of Kriging with External Drift (KED) is based on the assumption that the target variable (soil moisture) can be predicted from usually a linear function of the covariates. From our analysis in Section 3.2, such a prediction model does not exist, at

least on the whole experiment area. If local prediction model should be considered, the KED algorithm could be very complicated; and we are not sure we can fairly present the best result of KED. So, we do not include comparison with KED in the revised manuscript.

Minor technical corrections:

Minor technical corrections (1): What is the reason for the tilted perspective or the masking of the borders in the maps of Figs. 6 and 7? Simple two-dimensional plots might be a better solution. I recommend preparing Fig. 1 in a consistent way, i.e. use the same masking. What is the background colour shading in Fig. 1?

ANS:

The borders in maps of Figs. 6 and 7 of the old manuscript are border of the foci experiment field in the HiWater campaing. The reasons for masking the interpolation results with this border are as follows: (1) WSN nodes in the study area distributed within this border; (2) there are no land surface temperature data outside this border. Thus, we finally used the same masking for all the interpolation results.

The image shown in Fig. 1 was obtained from HJ satellite, combined by band 3(RED), band 4(NIR) and band2(GREEN). It has been masked with the same border in the revised manuscript.

Minor technical corrections (2): The term uncertainty analysis refers usually only to evaluations regarding the kriging standard deviation. I suggest renaming section 4.3 to cross validation and uncertainty analysis.

ANS:

Thanks for pointing out the concept differences. We have renamed this section (5.3 in the revised manuscript) title.

Minor technical corrections (3): Page 3, line 30 and other occurrences: The correct

spelling is cokriging.

ANS:

We have corrected this spelling mistake in the whole paper as the Cokriging.

Minor technical corrections (4): Page 7, lines 5-6: Why is the spherical model in particular important for structural and spatial interpolation? I recommend removing this clause.

ANS:

We have revised the concerned sentences, and removed the improper clause.

Please also note the supplement to this comment:
http://www.hydrol-earth-syst-sci-discuss.net/hess-2016-401/hess-2016-401-AC2-supplement.pdf
* * *
[Figure]

**Fig. 1.** The actual number of points considered for each interpolation day

**Supplement:**

[revised manuscript text omitted]
. Besides, when the soil moisture is close to saturation, the soil moisture sensors cannot work properly and sometimes give abnormal values (Zhang et al., 2015a). Thus, this part of data also need to be removed. We first excluded the abnormal data by assigning the zero value, negative value and abnormally high value (soil moisture content > 50 percent) as invalid data NaN. Then, we averaged the data for the whole day, and used the average result as the final soil moisture value for each node.

Remote sensing data served as auxiliary data in estimation of the distribution of near-surface soil moisture. We used both satellite remote sensing images and air borne remote sensing images to derive spectral variables. NDVI and albedo were derived from CCD camera on the Chinese HJ satellite, which has four channels in the visible and near infrared spectral range.

The spatial resolution was 30 m and revisiting frequency approximately was 2 days. Owing to the influence of clouds, only five clear-sky images on the following dates during this period could be used: June 15, June 19, June 29, July 8, and July 13 in the year 2012. All the images used here were pre-processed with calibration, geometric rectification, and atmospheric correction.

5     Land surface temperature (LST) is also  commonly used variable in monitoring soil moisture with remote sensing. The data were acquired from the airborne sensors of CASI/TASI on July 10, 2012, between 12:00 and 12:30 (local time), at an altitude of 2500 m (Xiao and Wen, 2013; Fan et al., 2015). The spatial resolution was 2.5 m. CASI acquires data covering the visible and near-infrared (VNIR) region of the spectrum. LST was derived from TASI, which is a hyperspectral thermal infrared sensor released by ITRES, Canada in 2006 (Wang et al., 2011). In this study, we introduced a soil moisture

10   map as simulation data in calculating the semivariogram and some interpolation analysis. This soil moisture map, which is presented in Fig. 7, was also derived from CASI/TASI data (Fan et al., 2015).

**3   Method**

The new spatial interpolation method proposed in this paper is based on the traditional Kriging algorithm. The proposed method extends the traditional X and Y spatial coordinates to spatial and spectral combined coordinates, and utilizes remote sensing

15   derived spectral variable NDVI and albedo in the interpolation algorithm as supplementary information. In this section, first, an outline of the traditional Kriging algorithm is given, then the technique employed to extend the Kriging algorithm is explained. While fitting the semivariogram of the soil moisture, as the number of WSN nodes is insufficient to gather robust statistics, we used the remote sensing-derived soil moisture map mentioned in Sect. 2.2 to calculate the variance function.

**3.1   Traditional Kriging method**

20   **3.1.1   Basic formula**

Kriging is an interpolation method derived from regionalized variable theory, which inherited the concept from geostatistics (Oliver and Webster, 1990). It has been used to provide linear unbiased predictions at unsampled locations and depends on expression of the spatial variation of the variable in terms of the semivariogram (Burgess and Webster, 1980; Cressie, 1990). This method quantifies and reduces the uncertainties of estimation, minimizing self-estimated prediction errors (Gao et al.,

25   2014). The core of Kriging is an optimally linear unbiased estimator that can be expressed as follows (Journel and Huijbregts, 1978):

$$Z^*(v_0) = \sum_{i=1}^{n} \lambda_i Z(v_i) \qquad (1)$$

where $Z^*$ is the estimated value of the variable at location $v_0$, $n$ is the number of the closest neighboring sampled data points used for interpolation, $\lambda_i$ is the Kriging weight assigned to each observation $Z(v_i)$.

Optimal estimation requires the minimum variance of errors:

$$\sigma_\kappa^2 \sigma_{\overline{\kappa}} = \text{Var}[Z(v_0) - Z^*(v_0)] = \text{E}\left\{\left[Z(v_0) - \sum_{i=1}^{n} \lambda_i Z(v_i)\right]^2\right\} = \min \qquad (2)$$

To ensure unbiased estimation, the following constraint must satisfy the equation as follows:

$$\sum_{i=1}^{n} \lambda_i = 1 \qquad (3)$$

5  To solve this constrained optimization problem, the Lagrange Multiplier Method (LMM) is adopted. With Eq. (2) as the objective function and Eq. (3) as the constraint, the LMM minimizes the following cost function:

$$f(\lambda_1, \lambda_2, \cdots, \lambda_n, \mu) = \frac{1}{2}\text{E}\left\{\left[Z(v_0) - \sum_{i=1}^{n} \lambda_i Z(v_i)\right]^2\right\} + \mu\left(1 - \sum_{i=1}^{n} \lambda_i\right) \qquad (4)$$

where $\mu$ is the Lagrange Multiplier. At the minimum point of the cost function, the differentiation of $f$ with respect to each of its variables is zero. Thus, the optimization problem decomposes into one of solving the following set of equations:

$$\begin{cases} \dfrac{\partial f}{\partial \lambda_i} = 0, & i = 1, 2, \cdots, n \\[2mm] \dfrac{\partial f}{\partial \mu} = 0 \end{cases} \qquad (5)$$

Differentiating the cost function, we have

$$f(\lambda_1, \lambda_2, \cdots, \lambda_n, \mu) = \frac{1}{2}\text{E}\left\{\left[Z(v_0) - \sum_{i=1}^{n} \lambda_i Z(v_i)\right]^2\right\} + \mu\left(1 - \sum_{i=1}^{n} \lambda_i\right) \qquad (6)$$

$$\begin{cases} \dfrac{\partial f}{\partial \lambda_i} = \lambda_i \text{E}[Z^2(v_i)] + \displaystyle\sum_{\substack{j=1 \\ j \neq i}}^{n} \lambda_j \text{E}[Z(v_i)Z(v_j)] - \text{E}[Z(v_0)Z(v_i)] - \mu = 0 \\[4mm] \dfrac{\partial f}{\partial \mu} = 1 - \displaystyle\sum_{i=1}^{n} \lambda_i = 0 \end{cases} \qquad (76)$$

If we know $\text{E}[Z^2(v_i)]$, $\text{E}[Z(v_0)Z(v_i)]$, and $\text{E}[Z(v_i)Z(v_j)]$, then the equations can be solved. These values are estimated by

15  the semivariogram function in Sect. 3.1.2.

The minimum variance of error ($\sigma_\kappa^2$), as is shown in Eq. (2), can be used as a quality indicator in estimation (Yamamoto, 2000). It can evaluate the intrinsic estimation uncertainty from the algorithm itself.

**3.1.2  Estimating semivariance and semivariogram**

20  Semivariance and semivariogram, containing spatial correlation information, are important concepts in geostatistics. The semivariance of variables at certain locations is estimated from the semivariogram function, which is a function of the distance between the two locations. Usually, a de-trending pre-process is applied to the observation data. After this pre-processing, the

spatial distribution of the variable is assumed stationary, which means that the semivariance does not change with location. On the basis of this assumption, the semivariance can be estimated from the data that a random variable is well correlated in space as a function of separation distance. The semivariance ($\gamma$) of Z between two data points is defined as

$$\gamma(x_i, x_0) = \gamma(h) = \frac{1}{2}\text{Var}[Z(x_i) - Z(x_0)] \qquad (7)$$

5    where $h$ is the distance between points $x_i$ and $x_0$, and $\gamma(h)$ is the semivariogram (Webster and Oliver, 2001).

The semivariogram is usually estimated from the statistics of sample points as follows:

$$\hat{\gamma}(h) = \frac{1}{2n}\sum_{i=1}^{n}\left(Z(x_i) - Z(x_i + h)\right)^2 \qquad (8)$$

where $n$ is the number of pairs of sample points separated by distance $h$ (Burrought and McDonnel, 1998).

As the number of WSN nodes is insufficient to gather robust statistics, the soil moisture map retrieved from airborne
10   hyperspectral remote sensing was used here to derive the semivariogram function.  In the calculation, we sampled  9000 random points  from the soil moisture map to calculate the semivariance  which is shown in Fig. 2. Here, we assumed that this semivariogram could be applied to interpolate WSN measured soil
15   moisture in the period from June 10, 2012 to July 15, 2012.

A spherical model was used in this study as the semivariogram model  (Burrought and McDonnel, 1998).

$$\gamma(h) = \begin{cases} C_0 + C_1\left[1.5*\left(\frac{h}{a}\right) - 0.5*\left(\frac{h}{a}\right)^3\right], & 0 \le h \le a \\ C_0 + C_1, & h > a \end{cases} \qquad (10)$$

20   where $\gamma(h)$ is the semivariance; $C_0$ represents a nugget, which is the minimum variability observed or the "noise" at a distance of zero; $C_1$ is the structural variance, $C_0 + C_1$ represents the sill variance; and $a$ is the range that signifies the correlation length in geostatistics.

The Kriging method requires the second-order stationarity for geostatistical inference and assumes it to be isotropic. As our study area was in the central part of the oasis, and no significant soil wetness spatial trend could be found, the soil moisture
25   observations in the study area were assumed to meet the above requirements. Because the area of the oasis is limited, i.e., the closest desert is about 4 km away from the center of the study area, the semivariogram statistics beyond 4 km may be affected by the presence of desert. Therefore, the range of the spherical model was set as 2500 m. Figure 2 shows the fitting curve of the semivariogram obtained using the spherical model. To reduce error, the spatial distance was divided into ten bins. We averaged the semivariance values of each bin as the final semivariance value of each distance. As the amount of the sampling
30   data was insufficient when the spatial distance was increased up to a certain extent, its average semivariance value was not

stable. Therefore, these invalid data were removed before semivariogram fitting. Hence, we only used the data when h<=4000 m in the fitting process of Fig. 2.

**3.2  Selection of spectral variables**

In this study, we selected NDVI and albedo as the auxiliary information to aid the interpolation. It is based on the following two main considerations: (1) NDVI and albedo are the spectral indexes which are fairly easy to be obtained from almost all high resolution remote sensing data sources; (2) NDVI and albedo represent most of the remote sensing information in the visible and near infrared spectral range. Although selecting NDVI and albedo as spectral variables is out of practical considerations, it is still necessary to analyze the correlation between soil moisture and NDVI/albedo, which are shown in Fig. 3.

In Fig. 3 (a) and (b), the NDVI and albedo data were collected from five available HJ satellites images to compare with the soil moisture data observed by WSN of the corresponding dates; in Fig.3 (c) and (d), the NDVI and albedo data were on July 13, comparing with the whole soil moisture inversion map after being masked. As can be seen from the comparison results, the NDVI and albedo are correlated to soil moisture to a certain extent, but the correlation coefficient is not significant (absolute value less than 0.5). An explanation of this result may be that NDVI can reflect the vegetation status, and vegetation usually grows good when soil moisture is abundant. However, in irrigated land, all the fields get enough irrigation; then the correlation between soil moisture and NDVI are weakened. In sparsely vegetated land, the albedo of dry soil is usually higher than that of wet soil. So, there is a correlation between soil moisture and albedo. But this correlation can be disturbed by soil type and the presence of vegetation. Actually, it is usually considered impossible to estimate soil moisture from visible and near infrared remote sensing data.

As land surface temperature map was acquired on July 10, 2012, with airborne remote sensing technique, we also analysed the correlation between soil moisture and LST. In order to get enough sample points, we use the soil moisture map from remote sensing inversion to correlate with LST, and the result is presented in Fig. 4. This analysis shows that LST has a better correlation with soil moisture (R=-0.53772) than that of NDVI and albedo, which explains why thermal infrared remote sensing is emphasised in drought monitoring. However, the correlation is not stable because of the complexity in land surface processes. It also should be mentioned that high resolution thermal infrared remote sensing image are currently rarely available data source, so, its potential can only benefit practical applications in the future.

**3.3  Extending the Kriging method to incorporate remote sensing information**

To reflect more details of the spatial distribution pattern of soil moisture, we propose a new algorithm that incorporates remote sensing variables, i.e., NDVI and albedo, into the basic Kriging method. The traditional interpolation space is the spatial space depicted by x and y coordinates. The new algorithm extends the interpolation space to the combined spatial and spectral space,

in which NDVI and albedo are treated as coordinates, just like $x$ and y. The distance in the combined space is characterized by the spatial distance and the spectral distance, as follows:

$$h = \sqrt{\Delta x^2 + \Delta y^2} \qquad (\sout{11}10)$$

$$s = \sqrt{(\frac{\Delta \text{NDVI}}{\sigma_{\text{DNVI}}})^2 + (\frac{\Delta \text{albedo}}{\sigma_{\text{albedo}}})^2} \qquad (\sout{12}11)$$

5 where $h$ is the spatial distance, $\Delta x$ and $\Delta y$ are the coordinate differences between two sampled points, $s$ represents the spectral distance, $\Delta$NDVI and $\Delta$albedo are the differences of NDVI and albedo values between two sampled points, and $\sigma_{\text{DNVI}}$ and $\sigma_{\text{albedo}}$ are two normalization factors (in this study, we simply set their values as 0.1, 0.1).

If LST is also used together with NDVI and albedo, the Eq. (11) will be expended as Eq. (12).

$$s = \sqrt{(\Delta \text{NDVI}/\sigma_{\text{DNVI}})^2 + (\Delta \text{albedo}/\sigma_{\text{albedo}})^2 + (\Delta \text{LST}/\sigma_{\text{LST}})^2} \quad (12)$$

10 where, $\Delta$LST represents the difference value of temperature between two sampled points from the airborne LST map, which is aggregated to 30 m resolution to match with NDVI and albedo, and $\sigma_{\text{LST}}$ is the normalization factors, the value of which is one degree.

Correspondingly, the semivariogram model was extended to the spatial and spectral combined space, as signified in the equations below:

$$\gamma_1(h) = \begin{cases} C_1 * \left[ 1.5 \left( \frac{h}{a_1} \right) - 0.5(\frac{h}{a_1})^3 \right], & 0 \le h \le a_1 \\ C_1, & h > a_1 \end{cases} \quad (13)$$

$$\gamma_2(s) = \begin{cases} C_2 * \left[ 1.5 \left( \frac{s}{a_2} \right) - 0.5(\frac{s}{a_2})^3 \right], & 0 \le s \le a_2 \\ C_2, & s > a_2 \end{cases} \quad (14)$$

$$\gamma(h,s) = \gamma_1(h) + \gamma_2(s) + C_0 \qquad (15)$$

where $\gamma_1$ and $\gamma_2$ are the semivariogram values with respect to $h$ and $s$, $\gamma$ is the overall semivariogram, and $a_1$ and $a_s$ are the lag distances of the spatial and spectral variables.

20   We also used the soil moisture map to derive the semivariance statistics. To reduce the error, the spatial/spectral distance was divided into ten bins as in that of traditional Kriging. The bin values with insufficient sample numbers were removed before semivariogram fitting. We only used the data when h<=4000 m and s<=4 in the fitting process. The semivariance, as a function of $h$ and $s$, is shown in Fig. 5.

   Here, the X-axis is spatial distance, the Y-axis is spectral distance, and the color in each grid represents the average
25 semivariance value of the soil moisture.

Using the semivariagram model as in the above Eq. (14), Eq. (15), and Eq. (16), the fitting semivariance diagram can be obtained, as shown in Fig. 6. The $a_1$ of spatial distance was preset as 2500 m, and the $a_2$ of spectral distance was preset as 3.5. Then, the fitted parameters $C_0$, $C_1$, and $C_2$ were 6.4468, 2.8762, and 2.9972, respectively.

**4   Analysis based on simulation data**

In this study, we used two or more auxiliary variables to aid the interpolation. Multivariate geostatistical technique algorithms, such as Cokriging, also provide various methods for combining auxiliary information. Here, we compared the proposed Extended Kriging with Cokring, using NDVI, albedo and land surface temperature as covariates. The Cokriging interpolation was implemented with the GSTAT package in R programming language.

In order to conduct this analysis and also ensure the amount of sampling points, we used the simulation data, the soil moisture map derived from airborne hyperspectral remote sensing, to extract sampling points and validation points. Firstly, as in Sect. 3.1.2, we sampled 9000 random points to calculate the semivariogram of Cokriging method. Then we used different numbers of points, ranging from 300 points to 30 points, to interpolate soil moisture. The numbers of sampling points are shown in Table 1. All the points in the soil moisture inversion map were used as validation points to calculate RMSE (Root Mean Square Error) and average $\sigma_k$, which is introduced in Sect. 3.1.1 as self assessment of interpolation uncertainty in Kriging algorithm, for each interpolation. As the locations of sampling points may influence the interpolation result and accuracy, we sampled randomly and repeated enough times to decrease the disturbing of point locations, and calculated the average RMSE value. For example, when the number of interpolation points is 30, we randomly sample 30 points for 100 times, interpolate respectively, and calculate the mean value of RMSE. The times of repetition and the interpolation accuracy comparison results are shown in Table 1.

From Table 1, we can see that the interpolation uncertainty (indicated both by $\sigma_\kappa$ and RMSE) decreases while the number of sampling points increases, and the estimator $\sigma_\kappa$ can reflect the variation trend of the actual RMSE. We also find out that the RMSE of Extended Kriging and Cokriging are very close, except that Cokriging performs a little better when the number of sample points is less than 50. The performance of Extended Kriging is similar with that of Cokriging in the quantitative perspective. However, the advantage of the Extended Kriging can be perceived when the interpolation results of the two methods are visually compared with the reference soil moisture map (Fig. 7). Figure 8 shows the interpolated soil moisture map by Extended Kriging and Cokring, both with the aid of NDVI and albedo: (a) is the result of Extended Kriging interpolated with 300 points; (b) is the result of Extended Kriging interpolated with 30 points; (c) is the result of Cokriging interpolated with 300 points; (d) is the result of Cokriging interpolated with 30 points. We can find that the interpolation results of Extended Kriging presented more detailed information for the spatial distribution of soil moisture than the results of Cokriging.

**45 Results and discussion**

[revised manuscript text omitted]

variations to soil moisture. However, there is a second explanation: because the remote sensing image was not available around the precipitation dates, we had to use an image that was acquired on an earlier date. For example, the interpolation result for June 28 is based on the image acquired on June 19; thus, this large uncertainty resulted from temporal mismatch between remote sensing data and ground data.

5

Land surface temperature (LST) is an important index connected with soil moisture, usually appearing as negatively correlated with soil moisture. Therefore, it is a potential spectral index that can support interpolation of soil moisture. However, high-resolution temperature remote sensing data are not as widely available as those of NDVI and albedo are. Fortunately, an LST map at 2.5 m resolution is available for July 10, 2012, as hyperspectral thermal infrared airborne images were acquired

10 by the TASI instrument on that date. Therefore, we added the LST as the third spectral index, together with NDVI and albedo, in the interpolation of this date, and compared the result with that obtained prior.  Estimation maps are shown in Fig. 15, where (a) is the result with the indexes of NDVI and albedo and (b) is the result with the indexes of NDVI, albedo, and LST. The comparison of uncertainty is shown in Table 2.

$$$$

15

The result indicates that introducing a new spectral temperature index can further improve the accuracy of soil moisture content: the value of $\sigma_k$ is reduced from 3.0343 to 2.8292 and the value of RMSE from 1.6406 to 1.3958. Hence, if high-resolution LST data are available in a long time series, the future  soil moisture interpolation task can incorporate the

20 LST information to boost accuracy.

**6 Conclusion**

With the rapid development of ground-based Earth observing techniques such as wireless sensor network, we are now able to monitor environmental parameters in real time, continuously, and with multiple sample points. However, interpolation is still needed to extend the point measurement to spatial distribution of the corresponding parameter in an area. As satellite remote

25 sensing is an efficient way of acquiring area Earth observing data, it is desirable to combine information from remote sensing and from ground-based observation networks.

The Extended Kriging method proposed in this study introduces the remote sensing image spectral information into the traditional interpolation method. NDVI and albedo are the spectral variables used in the algorithm. These spectral variables are treated in the same manner as the spatial variables, i.e., $x$ and $y$. Therefore, the interpolation is fundamentally the same

30 Kriging algorithm, but operating on the combined space of spatial dimension and spectral dimension. The semivariogram

model is also extended to the combined space. A remote sensing derived soil moisture map is used in this paper to fit the semivariogram model. However, this soil moisture map can be replaced by other sources of samples as long as the dataset is sufficiently large to derive robust statistics about the semivariance.

The proposed algorithm was applied to the soil moisture dataset acquire by the soil moisture sensors network
5    (WATERNET) in the oasis agricultural areas, which is the foci experimental area of the HiWATER campaign. As the WATERNET provides continuous near-surface soil moisture measurement over 48 scattered points, the interpolation results are daily soil moisture maps from June 10, 2012 to July 15, 2012, covering an area approximately 4.5 km $\times$ 5.0 km in size. Visual inspections indicate that the interpolation result from the proposed Extended Kriging algorithm presents much more spatial details than that of the traditional Kriging algorithm. The field-average soil moisture of several irrigation fields for long
10    time series are associated with the precipitation data and irrigation data, and the temporal variation of soil moisture can be well explained by these water inputs. The quantitative uncertainty analysis with both the leave-one-out method and $\sigma_k$ indicate that the Extended Kriging algorithm, which operates in the spectral and spatial combined space, produces more accurate interpolation results than that of the traditional Kriging algorithm, which operates only in the spatial domain. Currently, NDVI and albedo are recommended as the spectral variables to aid interpolation because they can be easily derived from most high-
15    resolution satellite images. However, we demonstrated in the discussion that more relevant spectral variables, such as land surface temperature, could be incorporated into this algorithm to improve its performance. However, how to choose the informative spectral variables remains an open topic for this algorithm.

There are other methods that can combine information from ground measurement and information from remote sensing, e.g., the Cokriging, the Kriging with External Drift, and the data assimilation method (Gao et al., 2014). Although
20    some results from the Cokriging are presented in Sect. 4, we prefer not to compare the proposed algorithm with other sophisticated algorithms in terms of accuracy for the following two considerations: in the first place, the Extended Kriging algorithm is much simpler than the Cokriging, the Kriging with External Drift, and the data assimilation method. Thus, it is possibly applicable in situations where the pre-conditions of other sophisticated algorithms are not satisfied. Second, as a new algorithm, the Extended Kriging

[revised manuscript text omitted]

Figures:

[Figure]

Figure 1 Study area in the middle reaches of the Heihe River Basin: the image was obtained from HJ satellite, combined by band 3 (RED), band 4 (NIR) and band2 (GREEN), and the black flags indicate the location of WATERNET nodes

[Figure]

Figure 2 Semivariance of sampled soil moisture data and fitting curve of the soil moisture semivariogram: the semivariance was calculated by the 9000 random sampling points from the soil moisture map; the semivariogram was fitted by spherical model with the data of h<=4000 m, and the range was set as 2500 m

Figure 3 the correlation analysis between auxiliary variables (NDVI and albedo) and soil moisture: (a) the NDVI data was derived from the five available HJ satellites images and compared with the soil moisture data observed by WSN of the corresponding dates; (b) the albedo data was derived from the five available HJ satellites images and compared with the soil moisture data observed by WSN of the corresponding dates; (c) the NDVI data was on July 13, and compared with the whole soil moisture inversion map on July 10, 2012; (d) the albedo data were on July 13, and compared with the whole soil moisture inversion map on July 10, 2012

[Figure]

Figure 4 the correlation analysis between LST and soil moisture: the soil moisture data were from the soil moisture inversion map, and the temperature data was derived from CASI/TASI on July 10, 2012

[Figure]

Figure 5 Semivariance of sampled soil moisture data with respect to spatial and spectral distance: the X-axis is spatial distance, the Y-axis is spectral distance, and the color in each grid represents the average semivariance value of the soil moisture; only valid semivariance data were kept for the semivariogram fitting (h<=4000 m and s<=4)

[Figure]

Figure 6 Fitting result for soil moisture semivariogram in the spatial and spectral dimensions: the X-axis is spatial distance, the Y-axis is spectral distance, and the color in each grid represents the average semivariance value of the soil moisture

[Figure]

5    Figure 7 the soil moisture inversion map used as the simulated data, derived from CASI/TASI on July 10, 2012

[Figure]

Figure 8 the estimation results interpolated by Extended Kriging and Cokring, both with the aid of NDVI and albedo: (a) is the result of Extended Kriging interpolated with 300 points; (b) is the result of Extended Kriging interpolated with 30 points; (c) is the result of Cokriging interpolated with 300 points; (d) is the result of Cokriging interpolated with 30 points.

[Figure]

Figure 9 Interpolated soil moisture at depths of 10 cm on July 10, 2012: (a) Interpolation result for traditional Ordinary Kriging method; (b) Interpolation result for Extended Kriging method

[Figure]

5    Figure 10  Interpolated soil moisture maps in the study area: the subfigures corresponds to the dates of June 10, June 17, June 24, July 1, July 8 and July 15, 2012, respectively

[Figure]

Figure 11 Location of the selected irrigation fields: A is WX-1, B is WX-4, C is XM-5, D is XM-9, E is WX-5, F is XM-3, G is SQ-2, H is KN-8, and I is SYZ-1

Figure 12 Comparison of soil moisture changing trend with precipitation and irrigation data of the chosen irrigation fields: the grey bar represents precipitation; the red arrow represents the irrigation; the blue line is the soil moisture calculated by Extended Kriging with the aid of NDVI and albedo; the grey dot line is the soil moisture calculated by the Ordinary Kriging

[Figure]

5  Figure 13 Comparison of the RMSE calculated by the traditional Ordinary Kriging and the Extended Kriging

[Figure]

Figure 14 Comparison of the $\sigma_k$ indicator calculated by the traditional Ordinary Kriging and the Extended Kriging

[Figure]

Figure 15 Soil moisture interpolation results at a depth of 10 cm on July 10, 2012: (a) Using the Extended Kriging method with the spectral indexes of NDVI and albedo; (b) using the Extended Kriging method with the spectral indexes of NDVI, albedo, and LST

5    Table:

Table 1 Comparison of the interpolation results for the Extended Kriging method and Cokriging method, using simulated data and the aid of auxiliary variables

| Number of points | Repetition times | NDVI and albedo | | | | NDVI, albedo and temperature | | | |
|---|---|---|---|---|---|---|---|---|---|
| | | Extended Kriging | | Cokrigng | | Extended Kriging | | Cokrigng | |
| | | $\sigma_\kappa$ | RMSE | $\sigma_\kappa$ | RMSE | $\sigma_\kappa$ | RMSE | $\sigma_\kappa$ | RMSE |
| 300 | 50 | 2.4451 | 2.3600 | 2.6888 | 2.3094 | 2.3745 | 2.2884 | 2.6861 | 2.3046 |
| 200 | 50 | 2.4921 | 2.4306 | 2.7481 | 2.3889 | 2.4133 | 2.3562 | 2.7449 | 2.3836 |
| 100 | 50 | 2.5927 | 2.5817 | 2.8722 | 2.5446 | 2.4949 | 2.5123 | 2.8686 | 2.5387 |
| 70 | 100 | 2.6564 | 2.7024 | 2.9560 | 2.6520 | 2.5450 | 2.6123 | 2.9467 | 2.6450 |
| 50 | 100 | 2.7214 | 2.8139 | 3.0316 | 2.7415 | 2.5951 | 2.7352 | 3.0276 | 2.7355 |
| 30 | 100 | 2.8317 | 3.0273 | 3.1749 | 2.9032 | 2.6777 | 2.9571 | 3.1710 | 2.8957 |

Table 2 Comparison of the interpolation results for the traditional Kriging method and the Extended Kriging method on July 10, 2012

| | Traditional Kriging | Extended Kriging with NDVI and albedo | Extended Kriging with NDVI, albedo and LST |
|---|---|---|---|
| $\sigma_k$ | 3.3125 | 3.0343 | 2.8292 |
| RMSE | 1.8937 | 1.6406 | 1.3958 |

---

## Author Comment (AC3) · 29 Nov 2016

Comments:

This paper introduces an interesting alternative model for prediction of a spatial variable by kriging. However, it does not reflect adequately on the motivation of the model, or explore its implications. There is therefore no real justification for the selection of the model, which strikes me as very implausible. Furthermore, the model is not implemented correctly. I expand on these statements below.

ANS:

Many thanks to the very long comments from Referee#2. We can see that Referee#2 is a real expert in geostatistics, and have made deep thinking to the proposed algorithm. All his comments are pertinent to the proposed method and very illuminating to us. We (the authors), however, are researchers in the field of remote sensing applications. We are single-mindedly trying to use remote sensing data to improve the estimation of soil moisture distribution. The emergence of WSN, as a new ground observation technique, brings not only potential benefit, but also the challenges to data quality control and the method on using these data. As we have known, soil moisture is an extremely heterogeneous environmental variable, very hard to be satisfactorily retrieved by remote sensing alone, or interpolated by Cokriging or even Kriging with External Drift algorithm with ground observations alone. The algorithm proposed in this manuscript is designed to suit the satellite remote sensing data and WSN observations, and it shows promising effect to some extent. Although we are quite aware that the proposed algorithm is not optimal and not even rigorous, we still wish to introduce it to the community, so that other experts, such as the anonymous referees, can help us to improve it. So, thanks very much to the referees and editors of HESS.

Comments:

First, there is no reason not to extend the geostatistical model from 1, 2 or 3 spatial dimensions to a space of higher dimensions. This is done, after all, in space-time geostatistics. However, if one is to treat some covariate as defining an additional dimension then, if one is assuming intrinsic stationarity in the new space, as one must for this extension of ordinary kriging, then this implies that there is no systematic relationship between the target variable and the covariate.

ANS:

It is true that the statistical correlation between soil moisture and the spectral variables is very weak. Following the suggestions of Referee#1, we have added some analysis on the soil moisture and the spectral variables into the manuscript (see Sect. 3.2 in the new manuscript). Figure 3 indicates that neither NDVI nor albedo has stable correlation with soil moisture (correlation coefficients less than 0.5). Land surface temperature

has a better correlation with soil moisture (Fig. 4), but it is not an easily available data source. I am not saying that there is no systematic relationship at all, but the relationship is too much complicated and unstable to use.

Comments:

In terms of the definition of the intrinsic hypothesis of stationarity E[Z(s)-Z(s+h)]=0 where Z(s) defines our random variable expressed as a random function of the covariate. In words, one is envisaging a situation in which a plot of the observations z against the corresponding values of the covariate s would not show any systematic trend but rather a random fluctuation, exhibiting some degree of correlation. This seems a rather implausible model to me. However, it might be worth considering.

ANS:

Actually, we should admit that more tests on the stationarity of soil moisture in the combined spatial and spectral space are needed. However, our data source is limited: the WSN observation nodes are scarce, and their data quality is not stable enough. As to the simulated data, i.e., the soil moisture map retrieved from airborne hyperspectral image, it is not the real soil moisture and its relation to the spectral variables could be misleading. Therefore, we can only ASSUME the random fluctuation and see the interpolation result. If the result is acceptable, we may try to collect more data to do the stationarity analysis later.

Comments:

Despite what the authors say, however, it is not a model one might use in a situation where cokriging or kriging with external drift were also candidate approaches, since in the case of kriging we assume a linear relationship between z and s, and in KED we assume a linear relationship or a relationship linear in the parameters of some simple fixed effects structure such as a polynomial or (non unduly complex) spline basis. This proposed approach and cokriging/KED would be fundamentally incompatible. Given

this, one would expect the authors so start by showing that cokriging or KED are not suitable for these data by plots and exploratory statistics that show (for cokriging) that a linear model of coregionalization is not plausible or (for KED) that no reasonable fixed effects structure looks reasonable.

ANS:

We agree with the Referee#2 that in situations when Cokriging or Kriging with External Drift is applicable, there is no advantage to use the proposed algorithm. However, to satisfy the condition of Cokriging or Kriging with External Drift is not easy to some extent, when it comes to soil moisture content. Following Referee#1's suggestion, we have added some interpolation result of Cokriging in Sect. 4 of the revised manuscript. The new figure (i.e., Fig. 8) indicates that due to the low correlation between soil moisture and spectral variables, the interpolation results of Cokriging are much too smooth, and the details of spatial pattern cannot be reflected. In comparison, the interpolation results of the proposed algorithm seem more alike the reference map (Fig.7). However, the RMSEs of Cokriging and the proposed algorithm are similar. The insignificant differences indicate that the advantage of the proposed algorithm is only in the visual aspect. This is understandable, because we can't expect the new algorithm to outperform the sophisticated Cokriging algorithm in many aspects.

Comments:

Let us assume that the authors do show a sound motivation for applying their model. They must then estimate it appropriately. I think there is a difficulty here. The authors should read the literature on space-time geostatistics to get a better understanding of this. One problem is that, unlike the space-time case, one cannot compute estimates of two marginal variograms (the spatial variogram with the time (or here s) lag =0, and vice versa). In this paper the authors cannot compute a marginal variogram in s space, because it is not possible to find observations with lag 0 in space but some non-zero lag in s. The variogram shown presumably uses lag bins, but the lack of the marginal

variogram is a problem because there is likely to be nugget variation in both dimensions and you cannot resolve their contribution to the joint space-s variogram.

ANS:

The referee really has thought deep into the problems of the new proposed algorithm. When we were designing this algorithm, we were handicapped by these problems, too. How to model the variogram in the combined spatial and spectral space? How to separate the variation caused by spatial distance and the variation caused by different spectral signature? And how to give weight to NDVI, albedo and temperature to properly construct the distance in spectral space? There are so many problems to solve. Then, we realized that we should not be trapped by these technical details and should not forget our initial motivation, either, i.e., introducing remote sensing information into the interpolation process, and finally pushing on the practical applications of WSN measurements. Hence, we deliberately chose the simplest solution to these problems and pushed on the overall algorithm. Our intention is, firstly, to introduce the overall ideal and the potential of the proposed algorithm; then the technical problems in the algorithm can be further studied in the future.

By the way, in estimating the variogram model (i.e., Eq. 15), we did not separately estimate the two marginal variograms. Instead, we simultaneously estimated the 3 model parameters (i.e., $C_0$, $C_1$, $C_2$) with surface fitting tools in MATLAB. The problem of "nugget variation in both dimensions" was avoided in this process.

Comments:

There is a second problem. The authors propose a simple model for the space-s variogram which is the sum of a spatial and an s-dimension variogram (Equation 15), but such a model is not in general valid (i.e. it does not define a non-negative definite covariance structure in the overall space). The authors should look at papers such as the one by De Cesare, L., Myers, D.E., Posa, D., 2001.[ Estimating and modeling space–time correlation structures. Stat. Probab. Lett. 51, 9–14.] for an account of this

and of some valid models.

ANS:

We admit that we have much to learn in geostatistics or space-time correlation structure. The reference paper which the referee recommended is very illuminating to us. However, for the current time, we cannot make too much change to the frame of this manuscript. On the other hand, we do not agree to the referee's judgment that our variogram model (Eq. 15 in the both version) does not define a non-negative definite covariance structure, actually it is non-negative, as is illustrated by Fig. 6.

Comments:

I have a further problem. The authors are using the variogram from a remote sensor for kriging from in-situ sensors. Even if they could be confident that the two variables are measurements of the same underlying quantity the variogram of the remote sensor data is a regularization of the variogram of the networked sensor data onto a very different spatial support. The question of how the remote sensor data might be used to help with this problem is interesting, given the fact that you have some data on the desired support it might be possible to do this, but it requires an explicit change of support step, not just ignoring the difference.

ANS:

Because of the limited number of in-situ sensors, we had to estimate variogram of soil moisture from remote sensing retrieved soil moisture map. The referee pointed out that the spatial support (or footprint) of the in-situ sensors may not be compatible with the support of pixels of the soil moisture map. This is a very good point. As soil moisture is a very heterogeneous variable, the measurement of in-situ sensors may only represent the average soil moisture in less than 1 square meter, but the pixel of the processed soil moisture map is 30*30 square meters to match with the resolution of HJ images. The scale difference between these two datasets is evident and may be an important

source of uncertainty.

Although a further investigation into this problem should be conducted, we found it impractical to lay too much stress on this topic in the current frame of paper. So, we only made some discussion of this problem in the conclusion part of the revised manuscript.

Please also note the supplement to this comment:
http://www.hydrol-earth-syst-sci-discuss.net/hess-2016-401/hess-2016-401-AC3-supplement.pdf

**Supplement:**

[revised manuscript text omitted]
. Besides, when the soil moisture is close to saturation, the soil moisture sensors cannot work properly and sometimes give abnormal values (Zhang et al., 2015a). Thus, this part of data also need to be removed. We first excluded the abnormal data by assigning the zero value, negative value and abnormally high value (soil moisture content > 50 percent) as invalid data NaN. Then, we averaged the data for the whole day, and used the average result as the final soil moisture value for each node.

Remote sensing data served as auxiliary data in estimation of the distribution of near-surface soil moisture. We used both satellite remote sensing images and air borne remote sensing images to derive spectral variables. NDVI and albedo were derived from CCD camera on the Chinese HJ satellite, which has four channels in the visible and near infrared spectral range.

The spatial resolution was 30 m and revisiting frequency approximately was 2 days. Owing to the influence of clouds, only five clear-sky images on the following dates during this period could be used: June 15, June 19, June 29, July 8, and July 13 in the year 2012. All the images used here were pre-processed with calibration, geometric rectification, and atmospheric correction.

Land surface temperature (LST) is also an important commonly used variable in monitoring soil moisture with remote sensing. The data were acquired from the airborne sensors of CASI/TASI on July 10, 2012, between 12:00 and 12:30 (local time), at an altitude of 2500 m (Xiao and Wen, 2013; Fan et al., 2015). The spatial resolution was 2.5 m. CASI acquires data covering the visible and near-infrared (VNIR) region of the spectrum. LST was derived from TASI, which is a hyperspectral thermal infrared sensor released by ITRES, Canada in 2006 (Wang et al., 2011). In this study, we introduced a soil moisture map as simulation data in calculating the semivariogram and some interpolation analysis. This soil moisture map, which is presented in Fig. 7, was also derived from CASI/TASI data (Fan et al., 2015).

**3 Method**

The new spatial interpolation method proposed in this paper is based on the traditional Kriging algorithm. The proposed method extends the traditional X and Y spatial coordinates to spatial and spectral combined coordinates, and utilizes remote sensing derived spectral variable NDVI and albedo in the interpolation algorithm as supplementary information. In this section, first, an outline of the traditional Kriging algorithm is given, then the technique employed to extend the Kriging algorithm is explained. While fitting the semivariogram of the soil moisture, as the number of WSN nodes is insufficient to gather robust statistics, we used the remote sensing-derived soil moisture map mentioned in Sect. 2.2 to calculate the variance function.

**3.1 Traditional Kriging method**

**3.1.1 Basic formula**

Kriging is an interpolation method derived from regionalized variable theory, which inherited the concept from geostatistics (Oliver and Webster, 1990). It has been used to provide linear unbiased predictions at unsampled locations and depends on expression of the spatial variation of the variable in terms of the semivariogram (Burgess and Webster, 1980; Cressie, 1990). This method quantifies and reduces the uncertainties of estimation, minimizing self-estimated prediction errors (Gao et al., 2014). The core of Kriging is an optimally linear unbiased estimator that can be expressed as follows (Journel and Huijbregts, 1978):

$$Z^*(v_0) = \sum_{i=1}^{n} \lambda_i Z(v_i) \quad (1)$$

where $Z^*$ is the estimated value of the variable at location $v_0$, $n$ is the number of the closest neighboring sampled data points used for interpolation, $\lambda_i$ is the Kriging weight assigned to each observation $Z(v_i)$.

Optimal estimation requires the minimum variance of errors:

$$\sigma_\kappa^2 \sigma_{\overline{\kappa}} = \mathrm{Var}[Z(v_0) - Z^*(v_0)] = \mathrm{E}\left\{\left[Z(v_0) - \sum_{i=1}^{n} \lambda_i Z(v_i)\right]^2\right\} = \min \qquad (2)$$

To ensure unbiased estimation, the following constraint must satisfy the equation as follows:

$$\sum_{i=1}^{n} \lambda_i = 1 \qquad (3)$$

5   To solve this constrained optimization problem, the Lagrange Multiplier Method (LMM) is adopted. With Eq. (2) as the objective function and Eq. (3) as the constraint, the LMM minimizes the following cost function:

$$f(\lambda_1, \lambda_2, \cdots, \lambda_n, \mu) = \frac{1}{2} \mathrm{E}\left\{\left[Z(v_0) - \sum_{i=1}^{n} \lambda_i Z(v_i)\right]^2\right\} + \mu\left(1 - \sum_{i=1}^{n} \lambda_i\right) \qquad (4)$$

where $\mu$ is the Lagrange Multiplier. At the minimum point of the cost function, the differentiation of $f$ with respect to each of its variables is zero. Thus, the optimization problem decomposes into one of solving the following set of equations:

$$\begin{cases} \dfrac{\partial f}{\partial \lambda_i} = 0, & i = 1, 2, \cdots, n \\ \dfrac{\partial f}{\partial \mu} = 0 \end{cases} \qquad (5)$$

Differentiating the cost function, we have

$$f(\lambda_1, \lambda_2, \cdots, \lambda_n, \mu) = \frac{1}{2} \mathrm{E}\left\{\left[Z(v_0) - \sum_{i=1}^{n} \lambda_i Z(v_i)\right]^2\right\} + \mu\left(1 - \sum_{i=1}^{n} \lambda_i\right) \qquad (6)$$

$$\begin{cases} \dfrac{\partial f}{\partial \lambda_i} = \lambda_i \mathrm{E}[Z^2(v_i)] + \displaystyle\sum_{\substack{j=1 \\ j \neq i}}^{n} \lambda_j \mathrm{E}\big[Z(v_i)Z(v_j)\big] - \mathrm{E}[Z(v_0)Z(v_i)] - \mu = 0 \\ \dfrac{\partial f}{\partial \mu} = 1 - \displaystyle\sum_{i=1}^{n} \lambda_i = 0 \end{cases} \qquad (76)$$

If we know $\mathrm{E}[Z^2(v_i)]$, $\mathrm{E}[Z(v_0)Z(v_i)]$, and $\mathrm{E}\big[Z(v_i)Z(v_j)\big]$, then the equations can be solved. These values are estimated by

15   the semivariogram function in Sect. 3.1.2.

The minimum variance of error ($\sigma_\kappa^2$), as is shown in Eq. (2), can be used as a quality indicator in estimation (Yamamoto, 2000). It can evaluate the intrinsic estimation uncertainty from the algorithm itself.

**3.1.2   Estimating semivariance and semivariogram**

20   Semivariance and semivariogram, containing spatial correlation information, are important concepts in geostatistics. The semivariance of variables at certain locations is estimated from the semivariogram function, which is a function of the distance between the two locations. Usually, a de-trending pre-process is applied to the observation data. After this pre-processing, the

spatial distribution of the variable is assumed stationary, which means that the semivariance does not change with location. On the basis of this assumption, the semivariance can be estimated from the data that a random variable is well correlated in space as a function of separation distance. The semivariance ($\gamma$) of Z between two data points is defined as

$$\gamma(x_i, x_0) = \gamma(h) = \frac{1}{2}\text{Var}[Z(x_i) - Z(x_0)] \qquad (\cancel{8}7)$$

5  where $h$ is the distance between points $x_i$ and $x_0$, and $\gamma(h)$ is the semivariogram (Webster and Oliver, 2001).

The semivariogram is usually estimated from the statistics of sample points as follows:

$$\hat{\gamma}(h) = \frac{1}{2n}\sum_{i=1}^{n}\left(Z(x_i) - Z(x_i + h)\right)^2 \qquad (\cancel{9}8)$$

where $n$ is the number of pairs of sample points separated by distance $h$ (Burrought and McDonnel, 1998).

As the number of WSN nodes is insufficient to gather robust statistics, the soil moisture map retrieved from airborne
10  hyperspectral remote sensing was used here to derive the semivariogram function.  In the calculation, we sampled  9000 random points  from the soil moisture map to calculate the semivariance  which is shown in Fig. 2. Here, we assumed that this semivariogram could be applied to interpolate WSN measured soil
15  moisture in the period from June 10, 2012 to July 15, 2012.

A spherical model was used in this study as the semivariogram model (Burrought and McDonnel, 1998).

$$\gamma(h) = \begin{cases} C_0 + C_1\left[1.5 * \left(\frac{h}{a}\right) - 0.5 * \left(\frac{h}{a}\right)^3\right], & 0 \leq h \leq a \\ C_0 + C_1, & h > a \end{cases} \qquad (\cancel{10}9)$$

20  where $\gamma(h)$ is the semivariance; $C_0$ represents a nugget, which is the minimum variability observed or the "noise" at a distance of zero; $C_1$ is the structural variance, $C_0 + C_1$ represents the sill variance; and $a$ is the range that signifies the correlation length in geostatistics.

The Kriging method requires the second-order stationarity for geostatistical inference and assumes it to be isotropic. As our study area was in the central part of the oasis, and no significant soil wetness spatial trend could be found, the soil moisture
25  observations in the study area were assumed to meet the above requirements. Because the area of the oasis is limited, i.e., the closest desert is about 4 km away from the center of the study area, the semivariogram statistics beyond 4 km may be affected by the presence of desert. Therefore, the range of the spherical model was set as 2500 m. Figure 2 shows the fitting curve of the semivariogram obtained using the spherical model. To reduce error, the spatial distance was divided into ten bins. We averaged the semivariance values of each bin as the final semivariance value of each distance. As the amount of the sampling
30  data was insufficient when the spatial distance was increased up to a certain extent, its average semivariance value was not

stable. Therefore, these invalid data were removed before semivariogram fitting. Hence, we only used the data when h<=4000 m in the fitting process of Fig. 2.

**3.2 Selection of spectral variables**

In this study, we selected NDVI and albedo as the auxiliary information to aid the interpolation. It is based on the following two main considerations: (1) NDVI and albedo are the spectral indexes which are fairly easy to be obtained from almost all high resolution remote sensing data sources; (2) NDVI and albedo represent most of the remote sensing information in the visible and near infrared spectral range. Although selecting NDVI and albedo as spectral variables is out of practical considerations, it is still necessary to analyze the correlation between soil moisture and NDVI/albedo, which are shown in Fig. 3.

In Fig. 3 (a) and (b), the NDVI and albedo data were collected from five available HJ satellites images to compare with the soil moisture data observed by WSN of the corresponding dates; in Fig.3 (c) and (d), the NDVI and albedo data were on July 13, comparing with the whole soil moisture inversion map after being masked. As can be seen from the comparison results, the NDVI and albedo are correlated to soil moisture to a certain extent, but the correlation coefficient is not significant (absolute value less than 0.5). An explanation of this result may be that NDVI can reflect the vegetation status, and vegetation usually grows good when soil moisture is abundant. However, in irrigated land, all the fields get enough irrigation; then the correlation between soil moisture and NDVI are weakened. In sparsely vegetated land, the albedo of dry soil is usually higher than that of wet soil. So, there is a correlation between soil moisture and albedo. But this correlation can be disturbed by soil type and the presence of vegetation. Actually, it is usually considered impossible to estimate soil moisture from visible and near infrared remote sensing data.

As land surface temperature map was acquired on July 10, 2012, with airborne remote sensing technique, we also analysed the correlation between soil moisture and LST. In order to get enough sample points, we use the soil moisture map from remote sensing inversion to correlate with LST, and the result is presented in Fig. 4. This analysis shows that LST has a better correlation with soil moisture (R=-0.53772) than that of NDVI and albedo, which explains why thermal infrared remote sensing is emphasised in drought monitoring. However, the correlation is not stable because of the complexity in land surface processes. It also should be mentioned that high resolution thermal infrared remote sensing image are currently rarely available data source, so, its potential can only benefit practical applications in the future.

**3.3 Extending the Kriging method to incorporate remote sensing information**

To reflect more details of the spatial distribution pattern of soil moisture, we propose a new algorithm that incorporates remote sensing variables, i.e., NDVI and albedo, into the basic Kriging method. The traditional interpolation space is the spatial space depicted by x and y coordinates. The new algorithm extends the interpolation space to the combined spatial and spectral space,

in which NDVI and albedo are treated as coordinates, just like $x$ and y. The distance in the combined space is characterized by the spatial distance and the spectral distance, as follows:

$$h = \sqrt{\Delta x^2 + \Delta y^2} \tag{10}$$

$$s = \sqrt{(\frac{\Delta \text{NDVI}}{\sigma_{\text{DNVI}}})^2 + (\frac{\Delta \text{albedo}}{\sigma_{\text{albedo}}})^2} \tag{11}$$

where $h$ is the spatial distance, $\Delta x$ and $\Delta y$ are the coordinate differences between two sampled points, $s$ represents the spectral distance, $\Delta$NDVI and $\Delta$albedo are the differences of NDVI and albedo values between two sampled points, and $\sigma_{\text{DNVI}}$ and $\sigma_{\text{albedo}}$ are two normalization factors (in this study, we simply set their values as 0.1, 0.1).

If LST is also used together with NDVI and albedo, the Eq. (11) will be expended as Eq. (12).

$$s = \sqrt{(\Delta \text{NDVI}/\sigma_{\text{DNVI}})^2 + (\Delta \text{albedo}/\sigma_{\text{albedo}})^2 + (\Delta \text{LST}/\sigma_{\text{LST}})^2} \tag{12}$$

where, $\Delta$LST represents the difference value of temperature between two sampled points from the airborne LST map, which is aggregated to 30 m resolution to match with NDVI and albedo, and $\sigma_{\text{LST}}$ is the normalization factors, the value of which is one degree.

Correspondingly, the semivariogram model was extended to the spatial and spectral combined space, as signified in the equations below:

$$\gamma_1(h) = \begin{cases} C_1 * \left[1.5 \left(\frac{h}{a_1}\right) - 0.5(\frac{h}{a_1})^3\right], & 0 \le h \le a_1 \\ C_1, & h > a_1 \end{cases} \tag{13}$$

$$\gamma_2(s) = \begin{cases} C_2 * \left[1.5 \left(\frac{s}{a_2}\right) - 0.5(\frac{s}{a_2})^3\right], & 0 \le s \le a_2 \\ C_2, & s > a_2 \end{cases} \tag{14}$$

$$\gamma(h,s) = \gamma_1(h) + \gamma_2(s) + C_0 \tag{15}$$

where $\gamma_1$ and $\gamma_2$ are the semivariogram values with respect to $h$ and $s$, $\gamma$ is the overall semivariogram, and $a_1$ and $a_s$ are the lag distances of the spatial and spectral variables.

We also used the soil moisture map to derive the semivariance statistics. To reduce the error, the spatial/spectral distance was divided into ten bins as in that of traditional Kriging. The bin values with insufficient sample numbers were removed before semivariogram fitting. We only used the data when h<=4000 m and s<=4 in the fitting process. The semivariance, as a function of $h$ and $s$, is shown in Fig. 5.

Here, the X-axis is spatial distance, the Y-axis is spectral distance, and the color in each grid represents the average semivariance value of the soil moisture.

When h > 4000 and s > 4, the sampled data quantity is not sufficient to satisfy statistical significance. Therefore, we divided h into six intervals ranging from 0 to 4000 m, and s into six intervals ranging from zero to four; the semivariance in Fig. 4 is the average value in these intervals.

Using the semivariagram model as in the above Eq. (13), Eq. (14), and Eq. (15), the fitting semivariance diagram can be obtained, as shown in Fig. 6. The $a_1$ of spatial distance was pre-set as 2500 m, and the $a_2$ of spectral distance was preset as 3.5. Then, the fitted parameters $C_0$, $C_1$, and $C_2$ were 6.4468, 2.8762, and 2.9972, respectively.

**4 Analysis based on simulation data**

In this study, we used two or more auxiliary variables to aid the interpolation. Multivariate geostatistical technique algorithms, such as Cokriging, also provide various methods for combining auxiliary information. Here, we compared the proposed Extended Kriging with Cokring, using NDVI, albedo and land surface temperature as covariates. The Cokriging interpolation was implemented with the GSTAT package in R programming language.

In order to conduct this analysis and also ensure the amount of sampling points, we used the simulation data, the soil moisture map derived from airborne hyperspectral remote sensing, to extract sampling points and validation points. Firstly, as in Sect. 3.1.2, we sampled 9000 random points to calculate the semivariogram of Cokriging method. Then we used different numbers of points, ranging from 300 points to 30 points, to interpolate soil moisture. The numbers of sampling points are shown in Table 1. All the points in the soil moisture inversion map were used as validation points to calculate RMSE (Root Mean Square Error) and average $\sigma_k$, which is introduced in Sect. 3.1.1 as self assessment of interpolation uncertainty in Kriging algorithm, for each interpolation. As the locations of sampling points may influence the interpolation result and accuracy, we sampled randomly and repeated enough times to decrease the disturbing of point locations, and calculated the average RMSE value. For example, when the number of interpolation points is 30, we randomly sample 30 points for 100 times, interpolate respectively, and calculate the mean value of RMSE. The times of repetition and the interpolation accuracy comparison results are shown in Table 1.

From Table 1, we can see that the interpolation uncertainty (indicated both by $\sigma_K$ and RMSE) decreases while the number of sampling points increases, and the estimator $\sigma_K$ can reflect the variation trend of the actual RMSE. We also find out that the RMSE of Extended Kriging and Cokriging are very close, except that Cokriging performs a little better when the number of sample points is less than 50. The performance of Extended Kriging is similar with that of Cokriging in the quantitative perspective. However, the advantage of the Extended Kriging can be perceived when the interpolation results of the two methods are visually compared with the reference soil moisture map (Fig. 7). Figure 8 shows the interpolated soil moisture map by Extended Kriging and Cokring, both with the aid of NDVI and albedo: (a) is the result of Extended Kriging interpolated with 300 points; (b) is the result of Extended Kriging interpolated with 30 points; (c) is the result of Cokriging interpolated with 300 points; (d) is the result of Cokriging interpolated with 30 points. We can find that the interpolation results of Extended Kriging presented more detailed information for the spatial distribution of soil moisture than the results of Cokriging.

**45 Results and discussion**

[revised manuscript text omitted]

variations to soil moisture. However, there is a second explanation: because the remote sensing image was not available around the precipitation dates, we had to use an image that was acquired on an earlier date. For example, the interpolation result for June 28 is based on the image acquired on June 19; thus, this large uncertainty resulted from temporal mismatch between remote sensing data and ground data.

5  ##

Land surface temperature (LST) is an important index connected with soil moisture, usually appearing as negatively correlated with soil moisture. Therefore, it is a potential spectral index that can support interpolation of soil moisture. However, high-resolution temperature remote sensing data are not as widely available as those of NDVI and albedo are. Fortunately, an LST map at 2.5 m resolution is available for July 10, 2012, as hyperspectral thermal infrared airborne images were acquired

10  by the TASI instrument on that date. Therefore, we added the LST as the third spectral index, together with NDVI and albedo, in the interpolation of this date, and compared the result with that obtained prior.  Estimation maps are shown in Fig. 15, where (a) is the result with the indexes of NDVI and albedo and (b) is the result with the indexes of NDVI, albedo, and LST. The comparison of uncertainty is shown in Table 2.

$$\sout{s \ = \ \sqrt{(\Delta NDVI/\sigma_{DNVI})^2 \ + \ (\Delta albedo/\sigma_{albedo})^2 \ + \ (\Delta LST/\sigma_{LST})^2} \quad (16)}$$

15

The result indicates that introducing a new spectral temperature index can further improve the accuracy of soil moisture content: the value of $\sigma_k$ is reduced from 3.0343 to 2.8292 and the value of RMSE from 1.6406 to 1.3958. Hence, if high-resolution LST data are available in a long time series, the future  soil moisture interpolation task can incorporate the

20  LST information to boost accuracy.

**6 Conclusion**

With the rapid development of ground-based Earth observing techniques such as wireless sensor network, we are now able to monitor environmental parameters in real time, continuously, and with multiple sample points. However, interpolation is still needed to extend the point measurement to spatial distribution of the corresponding parameter in an area. As satellite remote

25  sensing is an efficient way of acquiring area Earth observing data, it is desirable to combine information from remote sensing and from ground-based observation networks.

The Extended Kriging method proposed in this study introduces the remote sensing image spectral information into the traditional interpolation method. NDVI and albedo are the spectral variables used in the algorithm. These spectral variables are treated in the same manner as the spatial variables, i.e., $x$ and $y$. Therefore, the interpolation is fundamentally the same

30  Kriging algorithm, but operating on the combined space of spatial dimension and spectral dimension. The semivariogram

model is also extended to the combined space. A remote sensing derived soil moisture map is used in this paper to fit the semivariogram model. However, this soil moisture map can be replaced by other sources of samples as long as the dataset is sufficiently large to derive robust statistics about the semivariance.

5    The proposed algorithm was applied to the soil moisture dataset acquire by the soil moisture sensors network (WATERNET) in the oasis agricultural areas, which is the foci experimental area of the HiWATER campaign. As the WATERNET provides continuous near-surface soil moisture measurement over 48 scattered points, the interpolation results are daily soil moisture maps from June 10, 2012 to July 15, 2012, covering an area approximately 4.5 km $\times$ 5.0 km in size. Visual inspections indicate that the interpolation result from the proposed Extended Kriging algorithm presents much more spatial details than that of the traditional Kriging algorithm. The field-average soil moisture of several irrigation fields for long

10   time series are associated with the precipitation data and irrigation data, and the temporal variation of soil moisture can be well explained by these water inputs. The quantitative uncertainty analysis with both the leave-one-out method and $\sigma_k$ indicate that the Extended Kriging algorithm, which operates in the spectral and spatial combined space, produces more accurate interpolation results than that of the traditional Kriging algorithm, which operates only in the spatial domain. Currently, NDVI and albedo are recommended as the spectral variables to aid interpolation because they can be easily derived from most high-

15   resolution satellite images. However, we demonstrated in the discussion that more relevant spectral variables, such as land surface temperature, could be incorporated into this algorithm to improve its performance. However, how to choose the informative spectral variables remains an open topic for this algorithm.

There are other methods that can combine information from ground measurement and information from remote sensing, e.g., the Cokriging, the Kriging with External Drift, and the data assimilation method (Gao et al., 2014). Although

20   some results from the Cokriging are presented in Sect. 4, we prefer not to compare the proposed algorithm with other sophisticated algorithms in terms of accuracy for the following two considerations: in the first place, the Extended Kriging algorithm is much simpler than the Cokriging, the Kriging with External Drift, and the data assimilation method. Thus, it is possibly applicable in situations where the pre-conditions of other sophisticated algorithms are not satisfied. Second, as a new algorithm, the Extended Kriging

[revised manuscript text omitted]

Figures:

[Figure]

Figure 1 Study area in the middle reaches of the Heihe River Basin: the image was obtained from HJ satellite, combined by band 3 (RED), band 4 (NIR) and band2 (GREEN), and the black flags indicate the location of WATERNET nodes

[Figure]

Figure 2 Semivariance of sampled soil moisture data and fitting curve of the soil moisture semivariogram: the semivariance was calculated by the 9000 random sampling points from the soil moisture map; the semivariogram was fitted by spherical model with the data of h<=4000 m, and the range was set as 2500 m

Figure 3 the correlation analysis between auxiliary variables (NDVI and albedo) and soil moisture: (a) the NDVI data was derived from the five available HJ satellites images and compared with the soil moisture data observed by WSN of the corresponding dates; (b) the albedo data was derived from the five available HJ satellites images and compared with the soil moisture data observed by WSN of the corresponding dates; (c) the NDVI data was on July 13, and compared with the whole soil moisture inversion map on July 10, 2012; (d) the albedo data were on July 13, and compared with the whole soil moisture inversion map on July 10, 2012

[Figure]

Figure 4 the correlation analysis between LST and soil moisture: the soil moisture data were from the soil moisture inversion map, and the temperature data was derived from CASI/TASI on July 10, 2012

[Figure]

Figure 5 Semivariance of sampled soil moisture data with respect to spatial and spectral distance: the X-axis is spatial distance, the Y-axis is spectral distance, and the color in each grid represents the average semivariance value of the soil moisture; only valid semivariance data were kept for the semivariogram fitting (h<=4000 m and s<=4)

[Figure]

Figure 6 Fitting result for soil moisture semivariogram in the spatial and spectral dimensions: the X-axis is spatial distance, the Y-axis is spectral distance, and the color in each grid represents the average semivariance value of the soil moisture

[Figure]

5    Figure 7 the soil moisture inversion map used as the simulated data, derived from CASI/TASI on July 10, 2012

[Figure]

Figure 8 the estimation results interpolated by Extended Kriging and Cokring, both with the aid of NDVI and albedo: (a) is the result of Extended Kriging interpolated with 300 points; (b) is the result of Extended Kriging interpolated with 30 points; (c) is the result of Cokriging interpolated with 300 points; (d) is the result of Cokriging interpolated with 30 points.

[Figure]

Figure 9 Interpolated soil moisture at depths of 10 cm on July 10, 2012: (a) Interpolation result for traditional Ordinary Kriging method; (b) Interpolation result for Extended Kriging method

[Figure]

5   Figure 10  Interpolated soil moisture maps in the study area: the subfigures corresponds to the dates of June 10, June 17, June 24, July 1, July 8 and July 15, 2012, respectively

[Figure]

Figure 11 Location of the selected irrigation fields: A is WX-1, B is WX-4, C is XM-5, D is XM-9, E is WX-5, F is XM-3, G is SQ-2, H is KN-8, and I is SYZ-1

Figure 12 Comparison of soil moisture changing trend with precipitation and irrigation data of the chosen irrigation fields: the grey bar represents precipitation; the red arrow represents the irrigation; the blue line is the soil moisture calculated by Extended Kriging with the aid of NDVI and albedo; the grey dot line is the soil moisture calculated by the Ordinary Kriging

[Figure]

5    Figure 13 Comparison of the RMSE calculated by the traditional Ordinary Kriging and the Extended Kriging

[Figure]

Figure 14 Comparison of the $\sigma_k$ indicator calculated by the traditional Ordinary Kriging and the Extended Kriging

[Figure]

Figure 15 Soil moisture interpolation results at a depth of 10 cm on July 10, 2012: (a) Using the Extended Kriging method with the spectral indexes of NDVI and albedo; (b) using the Extended Kriging method with the spectral indexes of NDVI, albedo, and LST

5  Table:

Table 1 Comparison of the interpolation results for the Extended Kriging method and Cokriging method, using simulated data and the aid of auxiliary variables

| Number of points | Repetition times | NDVI and albedo | | | | NDVI, albedo and temperature | | | |
|---|---|---|---|---|---|---|---|---|---|
| | | Extended Kriging | | Cokrigng | | Extended Kriging | | Cokrigng | |
| | | $\sigma_\kappa$ | RMSE | $\sigma_\kappa$ | RMSE | $\sigma_\kappa$ | RMSE | $\sigma_\kappa$ | RMSE |
| 300 | 50 | 2.4451 | 2.3600 | 2.6888 | 2.3094 | 2.3745 | 2.2884 | 2.6861 | 2.3046 |
| 200 | 50 | 2.4921 | 2.4306 | 2.7481 | 2.3889 | 2.4133 | 2.3562 | 2.7449 | 2.3836 |
| 100 | 50 | 2.5927 | 2.5817 | 2.8722 | 2.5446 | 2.4949 | 2.5123 | 2.8686 | 2.5387 |
| 70 | 100 | 2.6564 | 2.7024 | 2.9560 | 2.6520 | 2.5450 | 2.6123 | 2.9467 | 2.6450 |
| 50 | 100 | 2.7214 | 2.8139 | 3.0316 | 2.7415 | 2.5951 | 2.7352 | 3.0276 | 2.7355 |
| 30 | 100 | 2.8317 | 3.0273 | 3.1749 | 2.9032 | 2.6777 | 2.9571 | 3.1710 | 2.8957 |

Table 2 Comparison of the interpolation results for the traditional Kriging method and the Extended Kriging method on July 10, 2012

| | Traditional Kriging | Extended Kriging with NDVI and albedo | Extended Kriging with NDVI, albedo and LST |
|---|---|---|---|
| $\sigma_k$ | 3.3125 | 3.0343 | 2.8292 |
| RMSE | 1.8937 | 1.6406 | 1.3958 |